# Fast Last-Iterate Convergence of SGD in the Smooth Interpolation Regime

**Amit Attia**[*]     **Matan Schliserman**[*]     **Uri Sherman**

Blavatnik School of Computer Science and AI

Tel Aviv University

{amitattia,schliserman,urisherman}@mail.tau.ac.il

**Tomer Koren**

Blavatnik School of Computer Science and AI

Tel Aviv University and Google Research

tkoren@tauex.tau.ac.il

## Abstract

We study population convergence guarantees of stochastic gradient descent (SGD) for smooth convex objectives in the interpolation regime, where the noise at optimum is zero or near zero. The behavior of the last iterate of SGD in this setting—particularly with large (constant) stepsizes—has received growing attention in recent years due to implications for the training of over-parameterized models, as well as to analyzing forgetting in continual learning and to understanding the convergence of the randomized Kaczmarz method for solving linear systems. We establish that after $T$ steps of SGD on $\beta$-smooth convex loss functions with stepsize $0 < \eta < 2/\beta$, the last iterate exhibits expected excess risk $\widetilde{O}\left(\frac{1}{\eta(2-\beta\eta)T^{1-\beta\eta/2}} + \frac{\eta}{(2-\beta\eta)^2}T^{\beta\eta/2}\sigma_\star^2\right)$, where $\sigma_\star^2$ denotes the variance of the stochastic gradients at the optimum. In particular, for a well-tuned stepsize we obtain a near optimal $\widetilde{O}(1/T + \sigma_\star/\sqrt{T})$ rate for the last iterate, extending the results of Varre et al. [2021] beyond least squares regression; and when $\sigma_\star = 0$ we obtain a rate of $O(1/\sqrt{T})$ with $\eta = 1/\beta$, improving upon the best-known $O(T^{-1/4})$ rate recently established by Evron et al. [2025] in the special case of realizable linear regression.

## 1 Introduction

We study the convergence of Stochastic Gradient Descent (SGD) for smooth convex objectives in the low-noise and interpolation regimes. Concretely, we consider optimization problems of the form

$$\min_{x \in \mathbb{R}^d} F(x) := \mathbb{E}_{z \sim \mathcal{Z}}[f(x; z)],$$

where $\mathcal{Z}$ is a distribution over a sample space $Z$, and $f(\cdot; z): \mathbb{R}^d \to \mathbb{R}$ is convex and $\beta$-smooth for all $z \in Z$. In the *low-noise regime*, the gradient noise at the optimum, defined as $\sigma_\star^2 := \mathbb{E}_z \|\nabla f(x^\star; z)\|^2$ for $x^\star \in \arg\min_{x \in \mathbb{R}^d} F(x)$, is assumed to be small or even zero. The special case where $\sigma_\star^2 = 0$ is known as the *interpolation regime*, in which $x^\star$ minimizes $f(\cdot, z)$ for almost all $z \in Z$. We seek bounds on the expected excess risk of SGD, that given an i.i.d. sample $z_1, \ldots, z_T \sim \mathcal{Z}$ performs the following updates, starting from an initialization $x_1 \in \mathbb{R}^d$:

$$x_{t+1} = x_t - \eta\nabla f(x_t; z_t). \qquad (t = 1, \ldots, T)$$

---

[*]*Equal contribution.*

39th Conference on Neural Information Processing Systems (NeurIPS 2025).

Standard convergence bounds of SGD apply to an average (possibly weighted) of the iterates. It is by now a classical result that in the smooth low-noise regime, the average iterate of SGD with a constant stepsize $\eta = \Theta(1/\beta)$ converges at a *fast rate* of $O(1/T + \sigma_\star/\sqrt{T})$ [Srebro et al., 2010, Ma et al., 2018], as compared to the slower $O(1/\sqrt{T})$ rate that holds more generally. In this work, we study *last-iterate bounds* for SGD, namely bounds that apply to the expected loss $\mathbb{E}[F(x_T)]$ of the last SGD iterate rather than to an average of the iterates. Somewhat surprisingly, despite a long line of work on last-iterate bounds [Rakhlin et al., 2012, Shamir and Zhang, 2013, Jain et al., 2019, Harvey et al., 2019, Varre et al., 2021, Zamani and Glineur, 2023, Liu and Zhou, 2024a, Evron et al., 2025], it has remained unknown whether fast rates in the smooth low-noise regime are also attained by the last iterate. Our goal in this paper is to fill in this gap.

There are several compelling reasons to study last-iterate bounds in low-noise regimes. First, as argued by Ma et al. [2018], the fast empirical convergence of SGD in training deep models, which is often run with a fixed constant stepsize, can be attributed to the fact that modern overparameterized networks are typically powerful enough to interpolate the data and reach zero loss, placing their optimization naturally in the interpolation regime. Second, it has been established that classical methods for solving ordinary least squares (OLS) problems and underdetermined systems of linear equations, such as the (randomized) Kaczmarz method, are instances of last-iterate SGD in the smooth interpolation regime [e.g., Needell et al., 2014]. Third, it has been recently demonstrated that last-iterate bounds for SGD in this regime are beneficial for analyzing catastrophic forgetting in a certain class of realizable continual learning problems [Evron et al., 2022, 2025], and this observation has led to improved analyses.

Crucially, however, some of the aforementioned applications of SGD in the interpolation regime hinge on a specific choice of a stepsize: $\eta = 1/\beta$. We refer to this as the "greedy stepsize" since, in the case of least squares (and in particular, in the Kaczmarz method), it leads to an update that steps directly to a minimizer of the instantaneous loss. Curiously though, previous analyses of SGD (either for the last or the average iterate) have not been able to treat the greedy choice $\eta = 1/\beta$ directly,[2] and have only provided bounds for stepsizes of the form $\eta = c/\beta$, for a constant $c$ strictly smaller than one [Srebro et al., 2010, Lan, 2012, Varre et al., 2021, Needell et al., 2014, Liu and Zhou, 2024a]. Thus, these analyses do not transfer to applications like continual learning and the randomized Kaczmarz method, which do require the exact setting of $\eta = 1/\beta$. Very recently, Evron et al. [2025] revisited this issue and established last-iterate bounds for SGD with $\eta = 1/\beta$ in the pure interpolation regime ($\sigma_\star = 0$), that scale as $O(1/T^{1/4})$ but hold only for OLS. Thus, there is still quite a significant gap between the best last-iterate bounds in this case and the fast $O(1/T)$ rate achieved by the average iterate of SGD with general smooth and convex losses.[3]

In this work, we provide the first (nearly-)optimal last-iterate convergence rate for SGD in the low-noise setting, significantly generalizing previous results limited to the OLS case in the pure interpolation regime with $\sigma_\star = 0$. Furthermore, we establish the first last-iterate guarantee with a constant "greedy" stepsize $\eta = 1/\beta$ in the interpolation regime and, as a consequence, substantially improve upon the best known rates for randomized Kaczmarz and realizable continual learning (in the condition-independent setting, not relying on strong convexity).

## 1.1 Summary of contributions

In more detail, in this paper we make the following contributions:

(i) We establish a convergence rate of

$$\widetilde{O}\left(\frac{1}{\eta(2 - \beta\eta)T^{1-\beta\eta/2}} + \frac{\eta}{(2 - \beta\eta)^2}T^{\beta\eta/2}\sigma_\star^2\right)$$

for the last iterate of $T$-steps SGD with stepsize $0 < \eta < 2/\beta$ on convex and $\beta$-smooth objectives, where $\sigma_\star^2$ is the variance of the stochastic gradients at the optimum. When $\eta \leq$

---

[2]While not stated explicitly in their paper, Bach and Moulines [2013] did provide a bound for $\eta = 1/\beta$, but only in the context of OLS and only for the average iterate of SGD.

[3]We remark that even for the average iterate, existing analyses do not directly apply to the greedy stepsize $\eta = 1/\beta$; we provide a more refined argument treating this case in Appendix B. The same applies to the last-iterate bounds in the strongly convex case derived in Needell et al. [2014], which also hold only for $\eta < 1/\beta$; we complement these results with an analysis for any stepsize $\eta < 2/\beta$, see details in Appendix C.

Table 1: **Convergence bounds for SGD in smooth convex settings.** The table considers the dependence on the number of iterations $T$, global noise bound $\sigma^2 \geq \sup_x \mathbb{E}\|\nabla f(x;z) - \nabla F(x)\|^2$, and variance at the optimum $\sigma_\star^2 \coloneqq \mathbb{E}\|\nabla f(x^\star;z) - \nabla F(x^\star)\|^2$. The step size of the algorithm is denoted by $\eta$. OLS stands for Ordinary Least Squares. Dependence on other parameters (e.g., the distance to the optimal solution), constants, and logarithmic factors is omitted for clarity. Only dimension-independent guarantees are included in the table.

| SETTING | REFERENCE | ADDITIONAL ASSUMPTIONS | OUTPUT ITERATE | CONVERGENCE RATE |
|---|---|---|---|---|
| $\beta$-smooth, convex $\eta$ optimally tuned | Lan [2012] | — | Average | $1/T + \sigma/\sqrt{T}$ |
| | Srebro et al. [2010][a] | — | Average | $1/T + \sigma_\star/\sqrt{T}$ |
| | Liu and Zhou [2024a] | — | Last | $1/T + \sigma/\sqrt{T}$ |
| | **This paper** | — | Last | $1/T + \sigma_\star/\sqrt{T}$ |
| $\beta$-smooth, convex realizable ($\sigma_\star = 0$) $\eta$ optimally tuned | Srebro et al. [2010] | — | Average | $1/T$ |
| | Varre et al. [2021] | OLS | Last | $1/T$ |
| | **This paper** | — | Last | $1/T$ |
| $\beta$-smooth, convex realizable ($\sigma_\star = 0$) $\eta = 1/\beta$ | Bach and Moulines [2013][b] | OLS | Average | $1/T$ |
| | Evron et al. [2025] | OLS | Last | $1/T^{1/4}$ |
| | **This paper** | — | Last | $1/\sqrt{T}$ |

[a] Srebro et al. [2010] established a slightly weaker guarantee that scales with the approximation error instead of $\sigma_\star$, but can be refined to $O(1/T + \sigma_\star/\sqrt{T})$.

[b] Bach and Moulines [2013] also established a convergence rate for ordinary least squares in the non-realizable setting; however, the result is dimension-dependent and therefore omitted from the table.

$1/(\beta \log T)$, our guarantee translates to $\widetilde{O}(1/(\eta T) + \eta \sigma_\star^2)$, providing the first fast last-iterate convergence rate in the low-noise regime (i.e., when $\sigma_\star \ll 1$). In particular, through an appropriate tuning of the stepsize we obtain a near optimal $\widetilde{O}(1/T + \sigma_\star/\sqrt{T})$ rate for the last iterate, extending the results of Varre et al. [2021] beyond least squares regression.

(ii) In the interpolation regime (i.e., when $\sigma_\star = 0$), we provide the first last-iterate convergence guarantee with "greedy" constant stepsize $\eta = 1/\beta$, achieving a rate of $O(1/\sqrt{T})$ and improving upon the best-known rate of $O(1/T^{1/4})$ in the special case of linear regression due to Evron et al. [2025]. This improved rate leads to better performance guarantees in applications such as continual linear regression and the randomized Kaczmarz method, via recent reductions to SGD with $\eta = 1/\beta$.

(iii) Finally, we extend our result in the interpolation regime to without-replacement SGD, achieving a similar $\widetilde{O}(1/T)$ fast rate and improving upon prior work where multiple passes over the samples were required for providing meaningful bounds [Cai and Diakonikolas, 2025, Liu and Zhou, 2024b].

See Table 1 for a summary of our results compared to existing art. We remark that while the improved $O(1/\sqrt{T})$ rate we establish in the greedy case $\eta = 1/\beta$ is likely not tight (we discuss optimality more below), it does *not* follow from the general $O(1/\sqrt{T})$ rate for the last iterate of SGD in the convex and Lipschitz (non-smooth) case [e.g., Shamir and Zhang, 2013, Liu and Zhou, 2024a]. Indeed, the latter convergence results require a non-constant stepsize of order $\eta = \widetilde{\Theta}(1/\sqrt{T})$, whereas our goal is to specifically address the particular choice of $\eta = 1/\beta$.

**Open problems.** Our work leaves several interesting questions for further investigation. Most notably, our rate in the regime where $\sigma_\star = 0$ and $\eta = 1/\beta$ is $O(1/\sqrt{T})$, whereas the best known lower bound in this setting is $\Omega(1/T)$ [e.g., Zhang et al., 2023], which also matches the upper bound for the average-iterate of SGD (see details in Appendix B). Closing this gap remains an open question for future work. As noted above, such an upper bound would also yield improved guarantees in continual learning and the Kaczmarz method. Notably, in those settings as well, the best known lower bound is $\Omega(1/T)$ [Evron et al., 2022]. Moreover, when the step size $\eta$ is optimally tuned, we obtain nearly optimal convergence rates up to logarithmic factors. Eliminating this remaining gap is left for future work. Finally, our convergence guarantee under without-replacement sampling currently holds

only in the interpolation regime. Whether this guarantee can be established in the low-noise regime remains an open question.

## 1.2 Related work

**Last-iterate analysis of SGD.** There is a rich line of work studying last-iterate convergence of SGD, most of which focuses on the convex Lipschitz setup in the gradient oracle model [Rakhlin et al., 2012, Shamir and Zhang, 2013, Jain et al., 2019, Harvey et al., 2019, Zamani and Glineur, 2023, Liu and Zhou, 2024a]. Surprisingly, this line of work has reached fruition with near-optimal bounds in all *but* the low-noise regimes. While the recent work of Liu and Zhou [2024a] additionally establishes last-iterate convergence for smooth convex objectives which may be applied in our setting, their bounds depend on a global noise parameter that can be arbitrarily larger than $\sigma_\star$. In particular, in the interpolation regime, their result translates to a suboptimal convergence rate of $O(1/\sqrt{T})$, whereas average-iterate bounds are known to achieve the faster rate of $O(1/T)$ [Srebro et al., 2010]. Moreover, their analysis does not establish convergence of SGD with the "greedy" stepsize $\eta = 1/\beta$, which is of particular importance due to the connection to continual learning and the Kaczmarz method [e.g., Needell et al., 2014, Evron et al., 2022, 2025]. By considering only the restricted class of ordinary least squares (OLS), several works have established last-iterate convergence results (see Table 1 for details). In contrast, we do not assume loss functions have any specific parametric form, nor do we assume the full objective is strongly convex.

**Overparametrization in deep learning.** Our work is additionally motivated by modern machine learning setups, where heavily over-parameterized deep neural networks are trained via SGD to perfectly fit the training data. Recent research connects over-parametrization with the ability to interpolate the training data, i.e., realizability of the empirical risk, and further, realizability with the fast convergence of SGD observed in practice [Ma et al., 2018]. In accordance, a growing line of work studies convergence behavior of SGD in the interpolation regime [e.g., Ge et al., 2019, Vaswani et al., 2019, Berthier et al., 2020, Varre et al., 2021, Wu et al., 2022, Liu et al., 2023], typically for under-determined linear regression problems, or under the assumption that component losses are convex and smooth and the full objective is strongly convex.

**The Kaczmarz method.** It is by now well-known that several variants of the (randomized) Kaczmarz method for solving underdetermined linear systems can be cast as instances of stochastic gradient descent [Needell et al., 2014]. Typical convergence bounds for the Kaczmarz method depend on the condition number of the coefficient matrix $A$; when the minimal singular value of $A$ is strictly positive, various variants of the method are known to converge linearly to the minimum-norm solution $x^\star$ [Gower and Richtárik, 2015, Han and Xie, 2024, Needell and Tropp, 2014]. However, the convergence rates in these settings can be arbitrarily slow, depending on the condition number of $A$, which is potentially arbitrarily large. Our results extend these findings by providing bounds that apply even when $\sigma_{\min} = 0$, independently of the condition number of $A$. Our guarantees in this case are in terms of the squared error, rather than of the Euclidean distance to a solution.

A key idea introduced by Strohmer and Vershynin [2009] in the context of the Kaczmarz method is to sample rows non-uniformly, with probabilities proportional to their squared norms. This sampling strategy fits naturally within the SGD framework, allowing us to analyze norm-based variants and derive a sharper bound (Corollary 4), where the bound depends on the average rather than the maximum row norm. In the block setting, both uniform and weighted sampling approaches have been explored [Needell and Tropp, 2014, Gower and Richtárik, 2015]. Similar to the single-row case, our results indicate that assigning sampling probabilities based on block norms can improve the analysis in a similar way, and imply better bounds through the average rather than the maximum block norm.

**Continual linear regression.** Prior work in continual learning has extensively studied the problem of catastrophic forgetting, demonstrating that the extent of forgetting is influenced by factors such as task similarity, model overparameterization, and the plasticity-stability trade-off [Goldfarb et al., 2024, Mermillod et al., 2013]. A growing body of theoretical and empirical research has shown that forgetting can diminish over time when tasks are presented in cyclic or random orderings [Evron et al., 2022, 2023, Kong et al., 2023, Jung et al., 2025, Cai and Diakonikolas, 2025, Lesort et al., 2023, Hemati et al., 2024, Evron et al., 2025]. Such orderings have been studied as mechanisms for mitigating forgetting, either through explicit control of learning environments or by modeling naturally

recurring patterns (e.g., seasonality in real-world domains). A recent line of work [Evron et al., 2022, 2025] demonstrated that convergence bounds for convex, $\beta$-smooth realizable optimization—such as those considered in our setting—can be applied to continual linear regression, where each task is a regression problem learned to convergence. Our more general results yield state-of-the-art bounds when applied in this setting. Concurrently to this work, Levinstein et al. [2025] showed that optimal $O(1/T)$ bounds in continual linear regression can also be achieved using a different approach: instead of fully optimizing each task, they present two methods—one that minimizes a regularized loss per task and another that performs a finite number of optimization steps.

**With vs. without-replacement sampling.** Our analysis of SGD without-replacement relates to prior work on average iterate convergence [e.g., Recht and Ré, 2012b, Nagaraj et al., 2019, Safran and Shamir, 2020, Rajput et al., 2020, Mishchenko et al., 2020, Cha et al., 2023, Cai et al., 2023]. For smooth, non-strongly convex objectives, near-optimal bounds have been established [Nagaraj et al., 2019, Mishchenko et al., 2020], with refined parameter dependence in the realizable case by Cai et al. [2023]. Last-iterate guarantees have more recently been studied by Liu and Zhou [2024b], Cai and Diakonikolas [2025], though these results assume non-constant step sizes and provide rates in terms of epochs, limiting their applicability to our setting. Specifically, in a realizable $\beta$-smooth setup, after $J$ without-replacement SGD epochs over a finite sum of size $n$, Mishchenko et al. [2020], Cai et al. [2023] obtained an $O(\beta/J)$ bound for the average iterate with step size $\eta = 1/(\beta n)$ and Liu and Zhou [2024b], Cai and Diakonikolas [2025] derived similar bounds for the last iterate up to logarithmic factors. In addition, Evron et al. [2025] recently analyzed the OLS setting with large step sizes, showing a last-iterate rate of $\widetilde{O}(1/T)$ for optimally tuned step sizes, and $\widetilde{O}(1/T^{1/4})$ for $\eta = 1/\beta$. We extend those results to general convex $\beta$-smooth realizable objectives, obtaining similar rates under optimal tuning and an improved $O(1/\sqrt{T})$ rate for $\eta = 1/\beta$ under without-replacement sampling.

The role of sampling order has also been studied in the context of the Kaczmarz method, where without-replacement sampling can lead to faster convergence. Recht and Ré [2012a] conjectured that a noncommutative arithmetic-geometric mean inequality could explain this advantage, but this was later disproven by Lai and Lim [2020], with further clarification by De Sa [2020]. Empirical work has shown that row-shuffling and cyclic orderings can perform comparably to i.i.d. sampling [Oswald and Zhou, 2015], aligning with broader observations in shuffled SGD [Bottou, 2009, Yun et al., 2021]. Our results support those of Evron et al. [2025] that suggests that with- and without-replacement sampling yield comparable performance up to constant factors. Thus, our analysis does not reveal a theoretical preference for one strategy over the other, leaving open the question of whether practical advantages can be gained through ordering.

**Concurrent work.** Following the initial publication of the present manuscript on arXiv [Attia et al., 2025], an independent work was released [Garrigos et al., 2025], which investigates the last-iterate convergence of SGD for convex and smooth objectives and establishes results closely related to ours. For a properly tuned stepsize, Garrigos et al. [2025] achieves the same nearly-optimal convergence rate of $\widetilde{O}(1/T + \sigma_\star/\sqrt{T})$. On the other hand, their result does not support for large stepsizes in the range $1/\beta \leq \eta < 2/\beta$ (in particular, it does not accommodate the "greedy" choice $\eta = 1/\beta$) and, compared with Theorem 2, provides weaker convergence rates for stepsizes of the form $\eta = c/\beta$ for a constant $c < 1$. In addition, the very recent work of Cortild et al. [2025], brought to our attention after the initial publication of our manuscript on arXiv, establishes *average-iterate* guarantees for SGD in the convex and smooth setting for any stepsize $\eta < 1/(2\beta)$, results that closely mirror those we include in Appendix B.

## 2   Last-iterate analysis in low-noise regimes

This section presents our main results concerning last-iterate convergence of SGD in the interpolation and low-noise regimes: in the interpolation regime, all functions share a common minimizer; while in the low-noise regime, the objective $F(x)$ admits a minimizer, and the variance of the stochastic gradients at that minimizer is small. Subsequently, we provide an overview of our analysis technique.

**Problem setup.** Let $Z$ be an index set for convex loss functions $f(\cdot; z)\colon \mathbb{R}^d \to \mathbb{R}$, and $\mathcal{Z} \in \Delta(Z)$ be an arbitrary distribution over $Z$. We consider the unconstrained stochastic optimization problem:

$$\min_{x \in \mathbb{R}^d} \{F(x) := \mathbb{E}_{z \sim \mathcal{Z}} f(x; z)\}, \tag{1}$$

under the assumption that the loss functions are individually smooth. Specifically, we assume that for all $z \in Z$, the function $f(\cdot; z)$ is $\beta$-smooth, namely that $\|\nabla f(y; z) - \nabla f(x; z)\| \le \beta \|y - x\|$ for all $x, y \in \mathbb{R}^d$. (Throughout, $\|\cdot\|$ refers to the $\ell_2$ norm.) We further assume that the objective $F$ admits a minimizer, denoted by $x^\star \in \arg\min_{x \in \mathbb{R}^d} F(x)$, and that the variance of the stochastic gradients at $x^\star$ is bounded, $\sigma_\star^2 := \mathbb{E}\|\nabla f(x^\star; z)\|^2 < \infty$.[4] The special case where $\sigma_\star^2 = 0$ is referred to as the *interpolation regime*, in which $x^\star$ minimizes $f(\cdot, z)$ for almost all $z \in Z$.

Throughout the paper, we consider the fixed stepsize SGD algorithm, which given an initialization $x_1 \in \mathbb{R}^d$, stepsize $\eta > 0$, number of steps $T$ and an i.i.d. sample $z_1, \ldots, z_T$, performs at each step $t = 1, \ldots, T$ the update

$$x_{t+1} = x_t - \eta \nabla f_t(x_t), \qquad \text{where } f_t(x) := f(x; z_t).$$

## 2.1 Main results

We next state our main results. We begin with the guarantee in the interpolation regime.

**Theorem 1** (last-iterate convergence in the interpolation regime). *Let $f\colon \mathbb{R}^d \times Z \to \mathbb{R}$ be such that $f(\cdot; z)$ is $\beta$-smooth and convex for every $z$. Assume there exists $x^\star \in \mathbb{R}^d$ such that $x^\star \in \arg\min_{x \in \mathbb{R}^d} f(x; z)$ for all $z \in Z$. Then, for SGD initialized at $x_1 \in \mathbb{R}^d$ with step size $\eta < 2/\beta$, where $T \ge 2$, we have the following last iterate guarantee:*

$$\mathbb{E}[F(x_T) - F(x^\star)] \le \frac{3\|x_1 - x^\star\|^2}{\eta(2 - \beta\eta)T^{1 - \beta\eta/2}}.$$

*In particular:*

(i) *when $\eta = \frac{1}{\beta}$, it holds that $\mathbb{E}[F(x_T) - F(x^\star)] \le \frac{3\beta\|x_1 - x^\star\|^2}{\sqrt{T}}$;*

(ii) *when $\eta = \frac{1}{\beta \log_2(T)}$, it holds that $\mathbb{E}[F(x_T) - F(x^\star)] \le \frac{6\beta\|x_1 - x^\star\|^2 \log_2(T)}{T}$.*

In comparison to the average-iterate guarantee of SGD [Srebro et al., 2010], the last-iterate guarantee in Theorem 1 incurs a suboptimal factor of $T^{\beta\eta/2}$; by choosing $\eta \le 1/(\beta \log_2 T)$, this suboptimality is reduced to a logarithmic factor and the bound nearly matches the average-iterate performance.

We further extend the above last-iterate convergence guarantee to the more general low-noise regime, where the noise at the optimum is small but nonzero.

**Theorem 2** (last-iterate convergence in the low-noise regime). *Let $f\colon \mathbb{R}^d \times Z \to \mathbb{R}$ be such that $f(\cdot; z)$ is $\beta$-smooth and convex for every $z$. Assume there exists a minimizer $x^\star \in \arg\min_{x \in \mathbb{R}^d} F(x)$, and that $\sigma_\star^2 < \infty$. Then, for $T$-steps SGD initialized at $x_1 \in \mathbb{R}^d$ with step size $\eta < 2/\beta$, where $T \ge 2$, we have the following last iterate guarantee:*

$$\mathbb{E}[F(x_T) - F(x^\star)] \le \frac{12\|x_1 - x^\star\|^2}{\eta(2 - \beta\eta)T^{1 - \beta\eta/2}} + \frac{24\eta}{(2 - \beta\eta)^2}\sigma_\star^2 T^{\beta\eta/2} \log_2(T + 2).$$

*In particular, when $\eta = \min\{1/\beta \log_2(T), \|x_1 - x^\star\|/\sqrt{\sigma_\star^2 T \log_2(T + 2)}\}$,*

$$\mathbb{E}[F(x_T) - F(x^\star)] \le \frac{24\beta\|x_1 - x^\star\|^2 \log_2(T)}{T} + \frac{120\sigma_\star\|x_1 - x^\star\|\sqrt{\log_2(T + 2)}}{\sqrt{T}}.$$

Theorem 2 is essentially an extension Theorem 1 (where we had $\sigma_\star = 0$), albeit with larger constant factors that stem from the discrepancy between $\|\nabla f(x_t; z)\|$ and $\|\nabla f(x_t; z) - \nabla f(x^\star; z)\|$ in the low-noise case, when $x^\star$ is not necessarily a minimizer of $f(x^\star; z)$.

---

[4]In case of multiple minima, $\sigma_\star$ can be defined with respect to any one of them—and in fact, its value can be shown to be the same across all minimizers; for additional details, see Appendix E.

## 2.2 Proof outline

Here we give an outline of the proof of Theorem 1, containing the main components of our analysis; for the full proof of Theorems 1 and 2, see Appendix G. While the outline is stated for the interpolation setting and with replacement sampling, the core components discussed also play a central role in our low-noise (Theorem 2) and sampling without-replacement (Theorem 3) results.

A natural starting point for obtaining last-iterate guarantees in the interpolation regime is through the analysis technique of Shamir and Zhang [2013], who provided the first last-iterate convergence guarantee in the general convex (non-smooth) setting. Recent previous work extended the technique to the interpolation regime [Evron et al., 2025], but was limited to linear regression and only achieved a suboptimal $O(1/T^{1/4})$ rate for $\eta = 1/\beta$.[5] An alternative approach, which was also taken by Liu and Zhou [2024a], is based on a powerful technique introduced by Zamani and Glineur [2023] for analyzing the subgradient method for deterministic (non-smooth) convex optimization, which they used for obtaining min-max optimal rates.

**Starting point: the Zamani & Glineur technique.** The starting point of our analysis is the technique of Zamani and Glineur [2023] for deriving last-iterate convergence bounds for the (deterministic) subgradient method in nonsmooth convex optimization. Starting at $x_1 \in \mathbb{R}^d$ and performing the deterministic update steps over a convex and differentiable function $F$ with a fixed stepsize, $x_{t+1} = x_t - \eta \nabla F(x_t)$ for $t = 1, 2, \ldots, T$, it follows from their analysis that

$$\eta \sum_{t=1}^{T} c_t (F(x_t) - F(x^\star)) \le \frac{v_0^2}{2} \|x_1 - x^\star\|^2 + \frac{\eta^2}{2} \sum_{t=1}^{T} v_t^2 \|\nabla F(x_t)\|^2,$$

where $c_t = v_t^2 - (v_t - v_{t-1}) \sum_{s=t}^{T} v_s$ and $0 < v_0 \le v_1 \le \cdots \le v_T$ are arbitrary non-decreasing weights. By carefully selecting the weights such that $c_T > 0$ and $c_t = 0$ for $1 \le t < T$, one can obtain a convergence bound for the last iterate $x_T$:

$$F(x_T) - F(x^\star) \le \frac{v_0^2}{2\eta c_T} \|x_1 - x^\star\|^2 + \frac{\eta}{2c_T} \sum_{t=1}^{T} v_t^2 \|\nabla F(x_t)\|^2.$$

**Step (i): from deterministic to stochastic smooth optimization.** Our first step is to generalize the analysis to the stochastic case, in which the update step at step $t$ is $x_{t+1} = x_t - \eta \nabla f(x_t; z_t)$, where $z_t \sim \mathcal{Z}$. Extending the analysis, we obtain that

$$\eta \sum_{t=1}^{T} c_t \mathbb{E}[F(x_t) - F(x^\star)] \le \frac{v_0^2}{2} \|x_1 - x^\star\|^2 + \frac{\eta^2}{2} \sum_{t=1}^{T} v_t^2 \mathbb{E}\|\nabla f_t(x_t)\|^2.$$

Using the inequality $\mathbb{E}\|\nabla f_t(x_t)\|^2 \le 2\beta \mathbb{E}[F(x_t) - F(x^\star)]$, which is a result of standard properties of convex and smooth functions, gives

$$\eta \sum_{t=1}^{T} \left(c_t - \beta\eta v_t^2\right) \mathbb{E}[F(x_t) - F(x^\star)] \le \frac{v_0^2}{2} \|x_1 - x^\star\|^2.$$

**Step (ii): modifying the weights $v_1, \ldots, v_T$.** The bound in the above display cannot yield a meaningful bound for $\eta = 1/\beta$, since the leading coefficient is negative: $c_t - v_t^2 < 0$. Even with $\eta < 1/\beta$, the previously specified weights which satisfy $c_t = 0$ for $1 \le t < T$ lead to a vacuous inequality. Using instead the weights setting $v_t = (T - t + 2)^{-\frac{1-\beta\eta}{2-\beta\eta}}$ for $t < T$ and $v_T = v_{T-1}$, for which one can show that $c_t - \beta\eta v_t^2 \ge 0$, we can establish that

$$\mathbb{E}[F(x_T) - F(x^\star)] = O\left(\beta \|x_1 - x^\star\|^2 T^{-1 + \frac{\beta\eta}{2-\beta\eta}}\right), \tag{2}$$

which is still vacuous for $\eta = 1/\beta$, but is strong enough for obtaining a rate of $\widetilde{O}(1/T)$ for stepsize $\eta = 1/\log_2(T)$.

---

[5]In Appendix D we provide a simplified analysis using the technique of Shamir and Zhang [2013] for the general convex and smooth setting; while the analysis achieves near-optimal guarantees for small stepsizes, it does not yield a guarantee for the choice $\eta = 1/\beta$ and provides weaker rates for large stepsizes $\eta = c/\beta$ (for a constant $c < 1$).

**Step (iii): tightening the regret analysis.** To further support the greedy setting $\eta = 1/\beta$ and to improve the above bound for general $\eta \leq 1/\beta$, we further tighten the analysis, obtaining

$$\eta \sum_{t=1}^{T} c_t \mathbb{E}[f_t(x_t) - f_t(x^\star)] \leq \frac{1}{2} v_0^2 \|x_1 - x^\star\|^2 + \frac{\eta^2}{2} \sum_{t=1}^{T} v_t^2 \mathbb{E}\|\nabla f_t(x_t)\|^2$$

$$- \frac{\eta}{2\beta} \sum_{t=1}^{T} v_t \left( \sum_{s=1}^{t} (v_s - v_{s-1}) \mathbb{E}\|\nabla f_t(x_t) - \nabla f_t(x_s)\|^2 + v_0 \mathbb{E}\|\nabla f_t(x_t)\|^2 \right).$$

This is achieved by replacing the use of the gradient inequality, $f(y) \geq f(x) + \langle \nabla f(x), y - x \rangle$, in the analysis of for general convex functions with the tighter lower bound for convex and smooth functions: $f(y) \geq f(x) + \langle \nabla f(x), y - x \rangle + \frac{1}{2\beta} \|\nabla f(y) - \nabla f(x)\|^2$.

**Step (iv): handling the cross terms.** A new challenge now arises from handling the cross terms $\|\nabla f_t(x_t) - \nabla f_t(x_s)\|^2$. A basic approach will be to use Young's inequality to bound

$$\|\nabla f_t(x_t) - \nabla f_t(x_s)\|^2 \leq (1 + \lambda)\|\nabla f_t(x_t)\|^2 + (1 + 1/\lambda)\|\nabla f_t(x_s)\|^2.$$

After rearranging the terms and carefully optimize over $\lambda$, one can already obtain a bound of

$$\mathbb{E}[F(x_T) - F(x^\star)] = O\left( \beta \|x_1 - x^\star\|^2 T^{-2 + \frac{1}{1 - \eta\beta/2 + \eta^2\beta^2/8}} \right),$$

achieving a rate of $O(T^{-2/5})$ with stepsize $\eta = 1/\beta$.

**Step (v): step-dependent Young's inequality.** We further tighten our analysis by carefully modifying the above Young's inequality parameter $\lambda$ for each $x_t, x_s$ pair. This leads us to the improved bound in Theorem 1, with a rate of $O(1/\sqrt{T})$. Additionally, for a stepsize of $\eta = 1/2\beta$, our improved analysis achieves a rate of $O(T^{-3/4})$, compared to the rate of $O(T^{-2/3})$ achieves by Eq. (2).

## 3 Extensions and implications

In this section, we show how our improved last-iterate convergence guarantees for SGD in the realizable setting, presented in Theorem 1, leads to sharper convergence rates in several key settings studied in the literature. In particular, we extend our analysis to without-replacement SGD, and derive new results in continual learning, leveraging the improved guarantees for both with and without-replacement sampling. These include the block Kaczmarz method, continual linear regression as well as continual linear classification and projection onto convex sets (for the latter, see Appendix A).

### 3.1 Without-replacement SGD

We first extend our result to a *without*-replacement sampling variant of the optimization problem Eq. (1). For this, we assume the instance set $Z$ is finite with $n := |Z|$, and consider the empirical risk minimization objective:

$$\min_{x \in \mathbb{R}^d} \left\{ F_Z(x) := \frac{1}{n} \sum_{z \in Z} f(x; z) \right\}, \tag{3}$$

which we will optimize by stochastic gradient descent w.r.t. a uniformly random shuffle of the training examples. For initialization $x_1 \in \mathbb{R}^d$, step size $\eta > 0$, without-replacement SGD samples a uniformly random permutation $\pi \sim \text{Unif}([n] \leftrightarrow Z)$, and iterates:

$$x_{t+1} \leftarrow x_t - \eta \nabla f(x_t; \pi_t). \tag{4}$$

We achieve the following convergence result,

**Theorem 3.** *Let $f : \mathbb{R}^d \times Z \to \mathbb{R}$ be such that $f(\cdot; z)$ is $\beta$-smooth and convex for every $z$. Assume further that there exists a joint minimizer $x^\star \in \cap_{z \in Z} \arg\min_{x \in \mathbb{R}^d} f(x; z)$. Then, for without-replacement SGD (Eq. (4)) initialized at $x_1 \in \mathbb{R}^d$ with step size $\eta < 2/\beta$, we have the following last iterate guarantee for all $2 \leq T \leq n$:*

$$\mathbb{E}[F_Z(x_T) - F_Z(x^\star)] \leq \frac{9\|x_1 - x^\star\|^2}{\eta(2 - \beta\eta)T^{1 - \beta\eta/2}} + \frac{4\beta^2\eta\|x_1 - x^\star\|^2}{T}.$$

*In particular, for $\eta = \frac{1}{\beta}$ we obtain $\mathbb{E}[F_Z(x_T) - F_Z(x^\star)] \leq 13\beta\|x_1 - x^\star\|^2/\sqrt{T}$, and for $\eta = \frac{1}{\beta \log_2 T}$ we obtain, $\mathbb{E}[F_Z(x_T) - F_Z(x^\star)] \leq 22\beta \log T\|x_1 - x^\star\|^2/T$.*

To prove Theorem 3, we first establish a bound on $\mathbb{E}[f_T(x_T) - f_T(x^\star)]$ as in the with-replacement case. We then apply the without-replacement algorithmic stability for smooth and realizable objectives provided by Evron et al. [2025]. The full proof appears in Appendix H.

### 3.2   The randomized (block-)Kaczmarz method

Block-Kaczmarz method is a classical algorithm for solving underdetermined linear systems of the form $Ax = b$ [Kaczmarz, 1937, Elfving, 1980]. The method initializes $x_1 = 0$, and at each iteration $t$, samples, with or without replacement, a random block $(A_{\tau(t)}, b_{\tau(t)})$ from the matrix $A$ and performs the update:

$$x_{t+1} \leftarrow x_t - cA_{\tau(t)}^+ \left(A_{\tau(t)}x_t - b_{\tau(t)}\right), \tag{5}$$

where the solution returned is the last iterate, $x_T$. The choice $c = 1$ corresponds to the standard Kaczmarz method, while values $c < 1$ are also of interest in certain settings (e.g., Needell et al., 2014).

In contrast to previous works (see discussion in Section 1.2), which consider the case where $A$ is a full-rank matrix and derives upper bounds on the distance between $x_T$ and the unique solution, $x^\star$, that depend on the condition number of $A$ (see Section 1.2 for a detailed discussion), our analysis evaluates the algorithm's performance using the average loss of the proposed solution, i.e., $F(x_T) = \frac{1}{2m}\sum_{j=1}^m \|A_j x_T - b_j\|^2$. Following is the guarantee for the block-Kaczmarz method.

**Corollary 4.** *Let $T \geq 2$, $c \leq 1$, and let $Ax = b$ be an equation system. Let $x^\star$ be such that $Ax^\star = b$. Assume that $\|A\|_2 \leq R$. Then, the block-Kaczmarz method (Eq. (5)) for $T$ iterations satisfies,*

$$\mathbb{E}\left[F(x_T)\right] = O\left(\frac{R^2\|x^\star\|^2}{T^{1-c\beta/2}}\right).$$

*In particular, for the standard method ($c = 1$) we obtain $\mathbb{E}[F(x_T) - F(x^\star)] \leq O(R^2\|x^\star\|^2/\sqrt{T})$, and for $c = 1/(\beta \log T)$ we obtain, $\mathbb{E}[F(x_T) - F(x^\star)] \leq \widetilde{O}(R^2\|x^\star\|^2/T)$.*

The bound given in Corollary 4 improves upon the result $O(1/T^{1/4})$ given by Evron et al. [2025] for the standard method ($c = 1$), and matches their bound when using the optimal choice $c = 1/\log T$. The full proof appears in Appendix I. We remark that even in the extensively studied well-conditioned case, the $c = 1$ case was not covered by existing results, as it corresponds to with SGD stepsize $\eta = 1/\beta$; we complement these results with a general analysis for any $0 < \eta < 2/\beta$, see details in Appendix C.

### 3.3   Continual linear regression

Continual linear regression has been studied extensively in recent years [e.g., Doan et al., 2021, Evron et al., 2022, Lin et al., 2023, Peng et al., 2023, Goldfarb and Hand, 2023, Li et al., 2023, Goldfarb et al., 2024, Hiratani, 2024, Evron et al., 2025]. In this setting, the learner is provided with a sequence of $m$ regression tasks, where each task is represented by a dataset $(A_j, b_j)$ for $j = 1, \ldots, m$, with $A_j \in \mathbb{R}^{n_j \times d}, b_j \in \mathbb{R}^{n_j}$.

The learner initializes with $x_1 = 0$ and, at each iteration, receives a task $(A_{\tau(t)}, b_{\tau(t)})$ that sampled uniformly at random (with or without replacement) and minimizes the squared loss for the current task by performing the update

$$x_{t+1} \leftarrow \left\{\text{minimize} \quad \|x - x_t\|^2 \quad \text{s.t.} \quad A_{\tau(t)}x = b_{\tau(t)}\right\}. \tag{6}$$

After $T$ iterations, the algorithm returns the final model $x_{T+1}$. We assume that $\|A_j\| \leq R$ for all $j \in [m]$, and we focus on the realizable case, where there exists a vector $x^\star \in \mathbb{R}^d$ such that $A_j x^\star = b_j$ for all $j \in [m]$. We consider two performance metrics: the *forgetting*, defined as $F_\tau(x_{T+1}) := \frac{1}{2T}\sum_{t=1}^T \|A_{\tau(t)}x_{T+1} - b_{\tau(t)}\|^2$, and the *population loss*, $F(x_{T+1}) := \frac{1}{2m}\sum_{j=1}^m \|A_j x_{T+1} - b_j\|^2$.[6]

---

[6]A more general definition of forgetting is $F_\tau(x_{T+1}) := \frac{1}{2T}\sum_{t=1}^T \|A_{\tau(t)}x_{T+1} - b_{\tau(t)}\|^2 - \frac{1}{2T}\sum_{t=1}^T \|A_{\tau(t)}x_{t+1} - b_{\tau(t)}\|^2$, but under Eq. (6) and realizability, the two definitions coincide.

Table 2: **Comparison of forgetting and loss rates for continual linear regression.** The same rates hold for block Kaczmarz with $c = 1$. Notation: $k$ = tasks, $T$ = iterations, $d$ = dimensionality, $\bar{r}$, $r_{\max}$ = average and maximum data-matrix ranks, $a \wedge b = \min(a, b)$. We omit multiplicative factors of $\|x^\star\|^2 R^2$.

| REFERENCE | BOUND | RANDOM ORDERING WITH REPLACEMENT | RANDOM ORDERING W/O REPLACEMENT |
|---|---|---|---|
| Evron et al. [2022] | Upper | $\dfrac{d - \bar{r}}{T}$ | — |
| Evron et al. [2025] | Upper | $\dfrac{1}{\sqrt[4]{T}} \wedge \dfrac{\sqrt{d - \bar{r}}}{T} \wedge \dfrac{\sqrt{m\,\bar{r}}}{T}$ | $\dfrac{1}{\sqrt[4]{T}} \wedge \dfrac{d - \bar{r}}{T}$ |
| **This paper** | Upper | $\dfrac{1}{\sqrt{T}}$ | $\dfrac{1}{\sqrt{T}}$ |
| Evron et al. [2022] | Lower | $\dfrac{1}{T}$ * | $\dfrac{1}{T}$ * |

\* Can be obtained by their construction for $m = 2$.

Previous work [Evron et al., 2022, 2023, 2025] has shown that the update rule in Eq. (6) can be reduced to an SGD update with step size $\eta = 1$ over modified loss functions. By leveraging this reduction and applying our main result for the last iterate of SGD (Theorem 1), we derive improved convergence guarantees for continual linear regression. In particular, using the convergence rate of SGD on $\beta$-smooth functions with step size $\eta = \frac{1}{\beta}$ from Theorem 1, we obtain the following result:

**Corollary 5.** *Suppose tasks are sampled uniformly at random, either with or without replacement, from a collection of m jointly realizable tasks. Then, after $T \geq 2$ iterations, the expected population loss and forgetting of the continual linear regression algorithm in Eq. (6) satisfy*

$$\mathbb{E}_\tau \left[ F\left(x_{T+1}\right) \right] = O\left( R^2 \|x^\star\|^2 / \sqrt{T} \right), \qquad \mathbb{E}_\tau \left[ F_\tau(x_{T+1}) \right] = O\left( R^2 \|x^\star\|^2 / \sqrt{T} \right).$$

The result in Corollary 5 improves the best known parameter-independent rates for continual linear regression from Evron et al. [2025] (see Table 2 for a more detailed comparison to previous work). For the proof of Corollary 5, see Appendix I.

### Acknowledgements

This project has received funding from the European Research Council (ERC) under the European Union's Horizon 2020 research and innovation program (grant agreement No. 101078075). Views and opinions expressed are however those of the author(s) only and do not necessarily reflect those of the European Union or the European Research Council. Neither the European Union nor the granting authority can be held responsible for them. This work received additional support from the Israel Science Foundation (ISF, grant numbers 2549/19 and 3174/23), a grant from the Tel Aviv University Center for AI and Data Science (TAD), from the Len Blavatnik and the Blavatnik Family foundation, from the Prof. Amnon Shashua and Mrs. Anat Ramaty Shashua Foundation, and a fellowship from the Israeli Council for Higher Education.

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

# A Additional extensions: Projection Onto Convex Sets (POCS) and continual linear classification

Here we provide additional extensions of our previous last iterate convergence result for SGD to other well-researched settings, Projection Onto Convex Sets (POCS) and continual linear classification. Comparison of our bounds in those regime to previous work appears in Table 3.

Table 3: **Forgetting Rates in Weakly-Regularized Continual Linear Classification on Separable Data.** We omit multiplicative factors of $\|x^\star\|^2 R^2$.

| Reference | Random with Replacement | Random w/o Replacement |
|---|---|---|
| Evron et al. [2023] | $\exp\left(-\frac{T}{4m\|w^\star\|^2 R^2}\right)$ | — |
| Evron et al. [2025] | $\frac{1}{\sqrt[4]{T}}$ | $\frac{1}{\sqrt[4]{T}}$ |
| **This paper** (2025) | $\frac{1}{\sqrt{T}}$ | $\frac{1}{\sqrt{T}}$ |

**Continual binary classification** is the setting of continual learning where the goal is to learn across $m \geq 2$ jointly separable tasks [e.g., Evron et al., 2025, 2023], where task $j$ is specified by a dataset

$$S_j = \{(z^{(i)}, y^{(i)})\}_{i=1}^{n_j}, \quad z^{(i)} \in \mathbb{R}^d, \ y^{(i)} \in \{-1, +1\},$$

and there exists a common separator $x^\star$ satisfying $y^{(i)} \langle z^{(i)}, x^\star \rangle \geq 1$ for every example in every task. At each iteration, the learner updates its weight vector by minimizing a weakly-regularized classification loss on the current task. Concretely, the procedure is:

---
**Algorithm 1** Regularized Continual Classification
---

Initialize $x_1 = 0$.
For each iteration $t = 1, \ldots, T$:
   Sample a dataset $S_t$ uniformly from $\{S_j\}_{j=1}^m$ (with or without replacement).

$$x_{t+1} \leftarrow \arg\min_{x \in \mathbb{R}^d} \sum_{(z,y) \in S_t} e^{-y x^T z} + \frac{\lambda}{2} \|x - x_t\|^2$$

Output $x_T$

---

The performance of the algorithm is measured by its forgetting, defined as,

$$F_\tau(x_T) = \frac{1}{2T} \sum_{t=1}^{T} \|x_T - \Pi_{\tau(t)}(x_T)\|.$$

We get the following result,

**Corollary 6.** *Under a random ordering, with or without replacement, over m jointly separable tasks, the expected forgetting of the weakly-regularized Algorithm 1 (at $\lambda \to 0$) after $T \geq 1$ iterations is bounded as*

$$\mathbb{E}_\tau\left[F_\tau(x_T)\right] \leq \frac{18\|x^\star\|^2 R^2}{\sqrt{T}}.$$

As discussed in Evron et al. [2023], when the magnitude of the regularization $\lambda$ goes to 0, this setting can be reduced for the setting of projections on convex sets, which is detailed below.

**Projection Onto Convex Sets (POCS)** is a method for solving a feasibility problem defined by $m$ closed convex sets $\{C_j\}_{j=1}^m$ (e.g Boyd et al. [2003], Gubin et al. [1967]). At each iteration $t = 1, \ldots, T$, we sample one constraint set index $\tau(t) \sim \text{Unif}([m])$ (with or without replacement) and project the current iterate $x_t$ onto the set $C_\tau(t)$. Namely,

---

**Algorithm 2** Projections onto Convex Sets (POCS)

---

1: **Initialize:** $x_1 = 0$
2: For $t = 1$ to $T$:
3:     Sample $\tau(t) \sim \text{Unif}([m])$.
4:     $x_{t+1} \leftarrow \Pi_{\tau(t)}(x_t) \triangleq \arg\min_{x \in C_{\tau(t)}} \|x - x_t\|$
5: Output $x_T$

---

We assume $\bigcap_{j=1}^m C_j \neq \emptyset$ (i.e. joint realizability), and measure performance by the average distance to projections on all sets:

$$F(w_T) = \frac{1}{2m} \sum_{j=1}^m \|w_T - \Pi_j(w_T)\|.$$

Using the reduction from SGD last-iterate bounds to sequential projections developed in Evron et al. [2025], we can plug in Theorem 1 and get the following results for continual linear classification and POCS,

**Corollary 7.** *Consider $m$ arbitrary (nonempty) closed convex sets $C_1, \ldots, C_m$, initial point $w_1 \in \mathbb{R}^d$ and assume a nonempty intersection $\bigcap_{j=1}^m C_j \neq \emptyset$. Then, under a random ordering with or without replacement, Algorithm 1 (and Algorithm 2) achieves,*

$$\mathbb{E}_\tau\left[\frac{1}{2m} \sum_{j=1}^m \|w_T - \Pi_j(w_T)\|^2\right] \leq \frac{7}{\sqrt{T}} \min_{x \in \cap C_j} \|x_1 - x\|^2.$$

The result given in Corollary 7, improves the best known parameter-independent rates for the loss continual linear classification and POCS method from Evron et al. [2025]. (see Table 3). The proof builds on our results for with and without-replacement SGD in the realizable setting (Theorems 1 and 3), combined with the reductions introduced by Evron et al. [2025, 2023],which relates last-iterate guarantees of SGD to performance guarantees in those regimes.

## A.1 Proof of Corollary 7

In the proof, we use the following lemmas from Evron et al. [2025].

**Lemma 1** (Reduction 3 in Evron et al. [2025]). *Consider $m$ arbitrary (nonempty) closed convex sets $C_1, \ldots, C_m$, initial point $x_1 \in \mathbb{R}^d$, and ordering $\tau$. Define $f_i(x) = \frac{1}{2}\|x - \Pi_i(x)\|^2, \forall i \in [m]$. Then,*

   *(i) $f_i$ is convex and 1-smooth.*

   *(ii) The POCS update is equivalent to an SGD step: $x_{t+1} = \Pi_{\tau(t)}(x_t) = x_t - \nabla_x f_{\tau(t)}(x_t)$.*

**Lemma 2** (From Proposition 17 in Evron et al. [2025]). *Under a random ordering $\tau$ without replacement, the iterates of Algorithm 2 (POCS) satisfy:*

$$\mathbb{E}_\tau\left[F(x_t)\right] = \frac{t}{T}\mathbb{E}_\tau\left[F_\tau(x_t)\right] + \frac{T-t}{2T}\mathbb{E}_\tau\left\|x_t - \Pi_{\tau(t)}(x_t)\right\|^2.$$

*Proof of Corollary 7.* Let $\tau$ be any random ordering, $x_1 \in \mathbb{R}^d$ an initialization, and $x_1, \ldots, x_T$ be the corresponding iterates produced by Algorithm 2. By Lemma 1, these are exactly the SGD iterates produced when initializing at $x_1$ and using a step size of $\eta = 1$, on the 1-smooth loss sequence $f_{\tau(1)}, \ldots, f_{\tau(k)}$ defined by:

$$f_i(x) := \frac{1}{2}\|x - \Pi_i(x))\|^2.$$

Since $f_i$ is convex and 1-smooth, we can apply Theorem 1 for $\eta = \frac{1}{\beta}$ and get, for the with-replacement case,

$$\mathbb{E}_\tau F(x_T) \leq \frac{3}{\sqrt{T}} \min_{x^\star \in \cap C} \|x_1 - x^\star\|^2,$$

where the minimum is since Theorem 1 is valid for any $x^\star \in \cap C$. For the without replacement case, by combining Theorem 3 and Lemma 2,

$$\mathbb{E}_\tau F(x_T) \leq \frac{7}{\sqrt{T}} \min_{x^\star \in \cap C} \|x_1 - x^\star\|^2.$$

$\square$

## A.2 Proof of Corollary 6

In the proof, we use the following lemma from Evron et al. [2025].

**Lemma 3** (Lemma 38 in Evron et al. [2025]). *Consider SGD with step size $\eta \leq 2/\beta$. For all $1 \leq T$, the following holds:*

$$\mathbb{E}F_\tau(x_T) \leq 2\mathbb{E}F(x_T) + \frac{4\beta^2\eta\|x_1 - x^\star\|^2}{T} .$$

*Proof of Corollary 6.* For the with-replacement case, the proof is implied by Corollary 7 and Lemma 3. For the without replacement, the proof is implied by Corollary 7 and Theorem 3. $\square$

## B Average-iterate convergence of SGD in the low-noise regime

In this section we provide an average-iterate convergence guarantee for SGD with a fixed stepsize, supporting any stepsize $\eta < 2/\beta$, and in particular, $\eta = 1/\beta$. The original analysis in this setting due to Srebro et al. [2010] only allowed for stepsizes $\eta < 1/\beta$; more recently, a more refined analysis was provided by Cortild et al. [2025] that supported large stepsizes $1/\beta \leq \eta < 2/\beta$, but at the price of a bias term that grows exponentially with $T$ when $\eta > 1/\beta$.

**Theorem 8.** *Let $f \colon \mathbb{R}^d \times Z \to \mathbb{R}$ be such that $f(\cdot; z)$ is $\beta$-smooth and convex for every $z$, $\mathcal{Z}$ a distribution over $Z$, and denote $F(x) := \mathbb{E}_{z \sim Z} f(x; z)$ . Assume there exists a minimizer $x^\star \in \arg\min_{x \in \mathbb{R}^d} F(x)$, and that $\sigma_\star^2 := \mathbb{E}_{z \sim Z}\|\nabla f(x^\star; z)\|^2 < \infty$. Then, for $T$-steps SGD initialized at $x_1 \in \mathbb{R}^d$ with step size $\eta < 2/\beta$, it holds that*

$$\mathbb{E}[F(\bar{x}) - F(x^\star)] \leq \frac{2\|x_1 - x^\star\|^2}{(2 - \beta\eta)\eta T} + \frac{8\eta\sigma_\star^2}{(2 - \beta\eta)^2},$$

*where $\bar{x} = \frac{1}{T} \sum_{t=1}^T x_t$. In particular, when $\eta = \min\{\frac{1}{\beta}, \frac{\|x_1 - x^\star\|}{\sqrt{4\sigma_\star^2 T}}\}$,*

$$\mathbb{E}[F(\bar{x}) - F(x^\star)] \leq \frac{2\beta\|x_1 - x^\star\|^2}{T} + \frac{8\|x_1 - x^\star\|\sigma_\star}{\sqrt{T}}.$$

*Proof.* We begin with a standard regret analysis. As $x_{t+1} = x_t - \eta\nabla f_t(x_t)$,

$$\|x_{t+1} - x^\star\|^2 = \|x_t - x^\star\|^2 - 2\eta\langle\nabla f_t(x_t), x_t - x^\star\rangle + \eta^2\|\nabla f_t(x_t)\|^2.$$

Rearranging, and summing for $t = 1, \ldots, T$,

$$\sum_{t=1}^T \langle\nabla f_t(x_t), x_t - x^\star\rangle = \frac{\|x_1 - x^\star\|^2 - \|x_{T+1} - x^\star\|^2}{2\eta} + \frac{\eta}{2}\sum_{t=1}^T\|\nabla f_t(x_t)\|^2.$$

Taking expectation and removing the $\|x_{T+1} - x^\star\|^2$ term,

$$\sum_{t=1}^T \mathbb{E}\langle\nabla f_t(x_t), x_t - x^\star\rangle \leq \frac{\|x_1 - x^\star\|^2}{2\eta} + \frac{\eta}{2}\sum_{t=1}^T \mathbb{E}\|\nabla f_t(x_t)\|^2.$$

By Young's inequality, for any $\lambda > 0$

$$\|\nabla f_t(x_t)\|^2 \le (1 + \lambda)\|\nabla f_t(x_t) - \nabla f_t(x^\star)\|^2 + (1 + 1/\lambda)\|\nabla f_t(x^\star)\|^2.$$

A standard property of a convex and $\beta$-smooth function $h$ [e.g., Nesterov, 1998] is that for any $x, y \in \mathbb{R}^d$,

$$\frac{1}{\beta}\|\nabla h(x) - \nabla h(y)\|^2 \le \langle \nabla h(x) - \nabla h(y), x - y \rangle.$$

Thus,

$$\|\nabla f_t(x_t)\|^2 \le \beta(1 + \lambda)\langle \nabla f_t(x_t) - \nabla f_t(x^\star), x_t - x^\star \rangle + (1 + 1/\lambda)\|\nabla f_t(x^\star)\|^2. \tag{7}$$

Taking expectation and noting that since $x^\star$ is a minimizer of $F(x)$, $\mathbb{E}\nabla f_t(x^\star) = 0$,

$$\mathbb{E}\|\nabla f_t(x_t)\|^2 \le \beta(1 + \lambda)\mathbb{E}\langle \nabla f_t(x_t), x_t - x^\star \rangle + (1 + 1/\lambda)\|\nabla f_t(x^\star)\|^2.$$

Plugging back to the regret analysis and rearranging,

$$\left(1 - \frac{\beta\eta(1 + \lambda)}{2}\right)\sum_{t=1}^{T} \mathbb{E}\langle \nabla f_t(x_t), x_t - x^\star \rangle \le \frac{\|x_1 - x^\star\|^2}{2\eta} + \frac{\eta(1 + 1/\lambda)}{2}\sum_{t=1}^{T} \mathbb{E}\|\nabla f_t(x^\star)\|^2.$$

Setting $\lambda = \frac{2 - \beta\eta}{2\beta\eta}$ and noting that $\mathbb{E}\|\nabla f_t(x^\star)\|^2 = \sigma_\star^2$,

$$\frac{1}{4}(2 - \beta\eta)\sum_{t=1}^{T} \mathbb{E}\langle \nabla f_t(x_t), x_t - x^\star \rangle \le \frac{\|x_1 - x^\star\|^2}{2\eta} + \frac{2 + \beta\eta}{2(2 - \beta\eta)}\eta\sigma_\star^2 T. \tag{8}$$

Thus, by a standard application of convexity and Jensen's inequality,

$$\mathbb{E}[F(\bar{x}) - F(x^\star)] \le \frac{2\|x_1 - x^\star\|^2}{(2 - \beta\eta)\eta T} + \frac{2(2 + \beta\eta)}{(2 - \beta\eta)^2}\eta\sigma_\star^2 \le \frac{2\|x_1 - x^\star\|^2}{(2 - \beta\eta)\eta T} + \frac{8\eta\sigma_\star^2}{(2 - \beta\eta)^2}.$$

In particular, setting $\eta = \min\{\frac{1}{\beta}, \frac{\|x_1 - x^\star\|}{\sqrt{4\sigma_\star^2 T}}\}$,

$$\mathbb{E}[F(\bar{x}) - F(x^\star)] \le \frac{2\beta\|x_1 - x^\star\|^2}{T} + \frac{8\|x_1 - x^\star\|\sigma_\star}{\sqrt{T}}. \qquad \square$$

## C   Last-iterate convergence of SGD in the strongly convex case

Here we show, for completeness, that in the low-noise strongly convex case, the last iterate of SGD converges at a linear rate for any stepsize $0 < \eta < 2/\beta$. Early results [Needell et al., 2014] achieved convergence only for stepsizes $\eta < 1/\beta$, and in particular, did not explicitly give bounds for the greedy choice $\eta = 1/\beta$. Cortild et al. [2025] has recently provided rates for any $0 < \eta < 2/\beta$, under the stronger assumption that $f(\cdot, z)$ is strongly convex for (almost all) all $z$ rather than in expectation. We remark that achieving last-iterate bounds (as opposed to average-iterate bounds) in the strongly convex case is entirely standard, and the only challenge here is to do so while accommodating large stepsizes.

**Theorem 9.** *Let $f: \mathbb{R}^d \times Z \to \mathbb{R}$ be such that $f(\cdot; z)$ is convex and $\beta$-smooth for every $z$, let $\mathcal{Z}$ a distribution over $Z$, and denote $F(x) := \mathbb{E}_{z \sim Z} f(x; z)$. Assume that $F$ is $\alpha$-strongly convex, minimized at $x^\star = \arg\min_{x \in \mathbb{R}^d} F(x)$, and that $\sigma_\star^2 := \mathbb{E}_{z \sim Z}\|\nabla f(x^\star; z)\|^2 < \infty$. Then, for $T$-step SGD initialized at $x_1 \in \mathbb{R}^d$ with stepsize $0 < \eta < 2/\beta$, it holds that*

$$\mathbb{E}\|x_{T+1} - x^\star\|^2 \le \exp\left(-\tfrac{1}{2}\eta(2 - \eta\beta)\alpha T\right)\|x_1 - x^\star\|^2 + \frac{8\eta\sigma_\star^2}{\alpha(2 - \eta\beta)^2}.$$

*In particular:*

*(i) for the greedy stepsize $\eta = 1/\beta$, we obtain:*

$$\mathbb{E}\|x_{T+1} - x^\star\|^2 \le \exp\left(-\frac{\alpha T}{2\beta}\right)\|x_1 - x^\star\|^2 + \frac{8\sigma_\star^2}{\alpha\beta};$$

(ii) *when $\sigma_\star^2 = 0$ (the interpolation regime), the bound is convergent for any $0 < \eta < 2/\beta$ and optimized when $\eta = 1/\beta$.*

*Proof.* We follow the standard analysis of SGD: since $x_{t+1} = x_t - \eta \nabla f_t(x_t)$,

$$\|x_{t+1} - x^\star\|^2 = \|x_t - x^\star\|^2 - 2\eta \langle \nabla f_t(x_t), x_t - x^\star \rangle + \eta^2 \|\nabla f_t(x_t)\|^2.$$

Using Eq. (7) we can bound, for any $\rho > 1$,

$$\|x_{t+1} - x^\star\|^2 \le \|x_t - x^\star\|^2 - 2\eta \langle \nabla f_t(x_t), x_t - x^\star \rangle$$
$$+ \eta^2 \left( \rho\beta \langle \nabla f_t(x_t) - \nabla f_t(x^\star), x_t - x^\star \rangle + \frac{\rho}{\rho - 1} \|\nabla f_t(x^\star)\|^2 \right).$$

Taking expectations, we obtain

$$\mathbb{E}\|x_{t+1} - x^\star\|^2 \le \mathbb{E}\|x_t - x^\star\|^2 - 2\eta \mathbb{E}\langle \nabla F(x_t), x_t - x^\star \rangle$$
$$+ \eta^2 \left( \rho\beta \mathbb{E}\langle \nabla F(x_t) - \nabla F(x^\star), x_t - x^\star \rangle + \frac{\rho}{\rho - 1} \sigma_\star^2 \right)$$
$$= \mathbb{E}\|x_t - x^\star\|^2 - \eta(2 - \rho\eta\beta) \mathbb{E}\langle \nabla F(x_t), x_t - x^\star \rangle + \frac{\rho}{\rho - 1} \eta^2 \sigma_\star^2.$$

On the other hand, since $F$ is $\alpha$-strongly convex, we have $\langle \nabla F(x_t), x_t - x^\star \rangle \ge \alpha \|x_t - x^\star\|^2$. We then obtain, whenever $2 - \rho\eta\beta > 0$, that

$$\mathbb{E}\|x_{t+1} - x^\star\|^2 \le \left(1 - \eta\alpha(2 - \rho\eta\beta)\right) \mathbb{E}\|x_t - x^\star\|^2 + \frac{\rho}{\rho - 1} \eta^2 \sigma_\star^2.$$

Now set $\rho = 1/(\eta\beta) + 1/2$ for which $1 < \rho < 2/(\eta\beta)$ as required, as $\eta < 2/\beta$. We obtain

$$\mathbb{E}\|x_{t+1} - x^\star\|^2 \le \left(1 - \tfrac{1}{2}\eta\alpha(2 - \eta\beta)\right) \mathbb{E}\|x_t - x^\star\|^2 + \frac{2 + \eta\beta}{2 - \eta\beta} \eta^2 \sigma_\star^2.$$

Unfolding the recursion results with

$$\mathbb{E}\|x_{T+1} - x^\star\|^2 \le \left(1 - \tfrac{1}{2}\eta\alpha(2 - \eta\beta)\right)^T \|x_1 - x^\star\|^2 + \frac{4\eta^2 \sigma_\star^2}{2 - \eta\beta} \sum_{t=0}^{\infty} \left(1 - \tfrac{1}{2}\eta\alpha(2 - \eta\beta)\right)^t$$

$$\le \exp\left(-\tfrac{1}{2}\eta(2 - \eta\beta)\alpha T\right) \|x_1 - x^\star\|^2 + \frac{8\eta\sigma_\star^2}{\alpha(2 - \eta\beta)^2}. \qquad \square$$

## D  Simpler last-iterate analysis for small stepsizes

Here we present an alternative analysis technique for the last-iterate convergence of SGD in the low-noise regime with small stepsizes $\eta \ll 1/\beta$, that in particular recovers the near-optimal rates in the regime $\eta \le 1/(\beta \log T)$ established in Theorem 2. The proof technique builds upon the earlier last-iterate approach of Shamir and Zhang [2013] and provides for a considerably simpler analysis; however, for large stepsizes (i.e., $\eta = \Omega(1/\beta)$), it yields inferior convergence rates as compared to the more intricate analysis of Theorem 2.

**Theorem 10.** *Let $f \colon \mathbb{R}^d \times Z \to \mathbb{R}$ be such that $f(\cdot; z)$ is $\beta$-smooth and convex for every $z$. Assume there exists a minimizer $x^\star \in \arg\min_{x \in \mathbb{R}^d} F(x)$, and that $\sigma_\star^2 < \infty$. Then, for $T$-steps SGD initialized at $x_1 \in \mathbb{R}^d$ with step size $0 < \eta < 1/\beta$, where $T \ge 3$, we have the following last iterate guarantee:*

$$\mathbb{E}[F(x_T) - F(x^\star)] \le \frac{16\|x_1 - x^\star\|^2}{\eta T^{1 - (2 - \eta\beta)\beta\eta}} + \frac{64\eta}{1 - \eta\beta} \sigma_\star^2 T^{(2 - \eta\beta)\beta\eta} \ln T.$$

*In particular, when $\eta = \min\left\{1/(\beta \ln(T)), \|x_1 - x^\star\|/\sqrt{\sigma_\star^2 T \ln(T)}\right\}$,*

$$\mathbb{E}[F(x_T) - F(x^\star)] \le \frac{144\beta \ln(T)\|x_1 - x^\star\|^2}{T} + \frac{720\sigma^* \|x_1 - x^\star\| \sqrt{\ln(T)}}{\sqrt{T}}.$$

Note that the bound above becomes non-convergent as $\eta\beta \to 1$, as opposed to the bound in Theorem 2, and in the interpolation regime it is dominated by the latter: e.g., for $\eta = 1/(2\beta)$ the bound above yields a rate of $T^{-1/4}$ compared to the $T^{-3/4}$ rate in Theorem 2.

*Proof.* Let $y \in \mathbb{R}^d$. Since $\langle \nabla f_t(x_t), x_t - y \rangle \geq f_t(x_t) - f_t(y)$ and, by Young's inequality, for every $\lambda > 0$, it holds that $\|\nabla f_t(x_t)\|^2 \leq (1+\lambda)\|\nabla f_t(x_t) - \nabla f_t(x^\star)\|^2 + (1+1/\lambda)\|\nabla f_t(x^\star)\|^2$,

$$\|x_{t+1} - y\|^2 = \|x_t - y\|^2 - 2\eta\langle \nabla f_t(x_t), x_t - y \rangle + \eta^2\|\nabla f_t(x_t)\|^2$$

$$\leq \|x_t - y\|^2 - 2\eta(f_t(x_t) - f_t(y)) + (1+\lambda)\eta^2\|\nabla f_t(x_t) - \nabla f_t(x^\star)\|^2 + \left(1 + \frac{1}{\lambda}\right)\eta^2\|\nabla f_t(x^\star)\|^2.$$

Taking expectation and noting that $\mathbb{E}\|\nabla f_t(x_t) - \nabla f_t(x^\star)\|^2 \leq 2\beta\mathbb{E}[f_t(x_t) - f_t(x^\star)]$, we get

$$\mathbb{E}\|x_{t+1} - y\|^2 \leq \mathbb{E}\|x_t - y\|^2 - 2\eta\mathbb{E}[f_t(x_t) - f_t(y)] + 2(1+\lambda)\beta\eta^2\mathbb{E}[f_t(x_t) - f_t(x^\star)] + \left(1 + \frac{1}{\lambda}\right)\eta\sigma_\star^2$$

$$= \mathbb{E}\|x_t - y\|^2 - 2\eta(1 - (1+\lambda)\beta\eta)\mathbb{E}[f_t(x_t) - f_t(y)]$$

$$+ 2(1+\lambda)\beta\eta^2\mathbb{E}[f_t(y) - f_t(x^\star)] + \left(1 + \frac{1}{\lambda}\right)\eta^2\sigma_\star^2,$$

where $\sigma_\star^2 := \mathbb{E}_z\|f(x^\star; z)\|^2$. Summing for $t = T - k, \ldots, T$,

$$\mathbb{E}\|x_{T+1} - y\|^2 \leq \mathbb{E}\|x_{T-k} - y\|^2 - 2\eta(1 - (1+\lambda)\beta\eta)\sum_{t=T-k}^{T}\mathbb{E}[f_t(x_t) - f_t(y)]$$

$$+ 2(1+\lambda)\beta\eta^2\sum_{t=T-k}^{T}\mathbb{E}[f_t(y) - f_t(x^\star)] + \left(1 + \frac{1}{\lambda}\right)\eta^2(k+1)\sigma_\star^2.$$

Setting $y = w_{T-k}$ and rearranging, we obtain, since $\mathbb{E}f_t(x_{T-k}) = \mathbb{E}f_{T-k}(x_{T-k})$ for $t \geq T - k$,

$$(1 - (1+\lambda)\eta\beta)\sum_{t=T-k}^{T}\mathbb{E}[F(x_t) - F(x^\star)] \leq (k+1)\mathbb{E}[F(x_{T-k}) - F(x^\star)] + \frac{1}{2}\left(1 + \frac{1}{\lambda}\right)\eta(k+1)\sigma_\star^2.$$

Now, denoting $S_k = \frac{1}{k+1}\sum_{t=T-k}^{T}\mathbb{E}[F(x_t) - F(x^\star)]$ and noticing that $\mathbb{E}[F(x_{T-k}) - F(x^\star)] = (k+1)S_k - kS_{k-1}$,

$$(1 - (1+\lambda)\eta\beta)S_k \leq (k+1)S_k - kS_{k-1} + \frac{1}{2}\left(1 + \frac{1}{\lambda}\right)\eta\sigma_\star^2.$$

Rearranging,

$$S_{k-1} \leq \left(1 + \frac{(1+\lambda)\beta\eta}{k}\right)S_k + \frac{\eta}{2k}\left(1 + \frac{1}{\lambda}\right)\sigma_\star^2.$$

Unfolding the recursion and setting $\lambda = 1 - \eta\beta$, we obtain

$$S_0 \leq S_{T-1}\prod_{k=1}^{T-1}\left(1 + \frac{(1+\lambda)\beta\eta}{k}\right) + \frac{\eta}{2}\left(1 + \frac{1}{\lambda}\right)\sigma_\star^2\sum_{k=1}^{T-1}\frac{1}{k}\prod_{s=1}^{k-1}\left(1 + \frac{(1+\lambda)\beta\eta}{s}\right)$$

$$\leq S_{T-1}\exp\left(\sum_{k=1}^{T-1}\frac{(1+\lambda)\beta\eta}{k}\right) + \frac{\eta}{2}\left(1 + \frac{1}{\lambda}\right)\sigma_\star^2\sum_{k=1}^{T-1}\frac{1}{k}\exp\left(\sum_{s=1}^{k-1}\left(\frac{(1+\lambda)\beta\eta}{s}\right)\right)$$

$$\leq S_{T-1}\exp((1+\lambda)\beta\eta\ln(eT)) + \frac{\eta}{2}\left(1 + \frac{1}{\lambda}\right)\sigma_\star^2\sum_{k=1}^{T}\frac{1}{k}\exp((1+\lambda)\beta\eta\ln(ek))$$

$$\leq \left(S_{T-1} + \frac{\eta}{2}\left(1 + \frac{1}{\lambda}\right)\sigma_\star^2\ln(eT)\right)\exp(2(1+\lambda)\beta\eta\ln(eT))$$

$$\leq \left(S_{T-1} + \eta\left(\frac{1}{1-\eta\beta}\right)\sigma_\star^2\ln(eT)\right)(eT)^{(2-\eta\beta)\beta\eta}, \qquad\qquad (\lambda = 1 - \eta\beta, \eta\beta < 1)$$

where the third and fourth inequalities follow by $\sum_{i=n}^{N-1} \frac{1}{n} \le 1 + \ln(N) = \ln(eN)$. By applying convexity and using the average-iterate convergence rate in Eq. (8), we know that

$$S_{T-1} = \frac{1}{T} \sum_{t=1}^{T} \mathbb{E}[F(x_t) - F(x^\star)] \le \frac{1}{T}(2 - \beta\eta) \sum_{t=1}^{T} \mathbb{E}\langle \nabla f_t(x_t), x_t - x^\star \rangle$$

$$\le \frac{2\|x_1 - x^\star\|^2}{\eta T} + \frac{2(2 + \beta\eta)}{2 - \beta\eta}\eta\sigma_\star^2 \le \frac{2\|x_1 - x^\star\|^2}{\eta T} + 6\eta\sigma_\star^2,$$

where the first and last inequalities use $\eta \le 1/\beta$. Then, we get

$$\mathbb{E}[F(x_T) - F(x^\star)] = S_0 \le \left( \frac{2\|x_1 - x^\star\|^2}{\eta T} + 6\eta\sigma_\star^2 + \frac{\eta}{1 - \eta\beta}\sigma_\star^2 \ln(eT) \right)(eT)^{(2 - \eta\beta)\beta\eta}$$

$$\le \frac{16\|x_1 - x^\star\|^2}{\eta T^{1 - (2 - \eta\beta)\beta\eta}} + \frac{64\eta}{1 - \eta\beta}\sigma_\star^2 \ln(T) T^{(2 - \eta\beta)\beta\eta}.$$

In particular, when $\eta = \min\left\{ 1/(\beta \ln(T)), \|x_1 - x^\star\|/\sqrt{\sigma_\star^2 T \ln(T)} \right\}$,

$$\mathbb{E}F(x_T) - F(x^\star) \le \frac{144\beta \ln(T)\|x_1 - x^\star\|^2}{T} + \frac{720\sigma^\star\|x_1 - x^\star\|\sqrt{\ln(T)}}{\sqrt{T}}. \qquad \square$$

## E  Multiple minimizers in the low-noise regime

In this section we show that assuming $f(\cdot; z)$ is convex and smooth for any $z \in Z$, the variance at the optimum $x^\star \in \arg\min_{x \in \mathbb{R}^d} F(x)$ (assuming one exists) is the same, no matter which optimum is selected. Following is the result.

**Theorem 11.** *Let $f: \mathbb{R}^d \times Z \to \mathbb{R}$ be such that $f(\cdot; z)$ is $\beta$-smooth and convex for every $z$, and $\mathcal{Z}$ be a distribution over $Z$. Assume there exists a minimizer $x_1^\star \in \arg\min_{x \in \mathbb{R}^d} F(x) \coloneqq \mathbb{E}_{z \sim \mathcal{Z}} f(x; z)$, and that $\mathbb{E}_{z \sim \mathcal{Z}}[\|\nabla f(x_1^\star; z)\|^2] < \infty$. Then for any $x_2^\star \in \arg\min_{x \in \mathbb{R}^d} F(x)$,*

$$\mathbb{E}_{z \sim \mathcal{Z}}[\|\nabla f(x_2^\star; z)\|^2] = \mathbb{E}_{z \sim \mathcal{Z}}[\|\nabla f(x_1^\star; z)\|^2].$$

*Proof.* A standard property of a convex and $\beta$-smooth function $h$ [e.g., Nesterov, 1998] is that for any $x, y \in \mathbb{R}^d$,

$$\frac{1}{\beta}\|\nabla h(x) - \nabla h(y)\|^2 \le \langle \nabla h(x) - \nabla h(y), x - y \rangle.$$

Thus,

$$\mathbb{E}\|\nabla f(x_1^\star; z) - \nabla f(x_2^\star; z)\|^2 \le \beta\mathbb{E}\langle \nabla f(x_1^\star; z) - \nabla f(x_2^\star; z), x_1^\star - x_2^\star \rangle$$
$$= \beta\langle \nabla F(x_1^\star), x_1^\star - x_2^\star \rangle - \beta\langle \nabla F(x_2^\star), x_1^\star - x_2^\star \rangle$$
$$= 0,$$

where the last equality follows since $x_1^\star, x_2^\star$ are minimizers of $F(x)$. The random variable $\|\nabla f(x_1^\star; z) - \nabla f(x_2^\star; z)\|^2$ is therefore non-negative and has expectation zero, implying that $\nabla f(x_1^\star; z) = \nabla f(x_2^\star; z)$ almost surely and thus concluding our proof. $\qquad \square$

## F  Technical lemmas

**Lemma 4.** *Let $c \ge 1$, $\alpha \in (0, 1)$, and $n \in \mathbb{N}$. Then*

$$(1 + c)^{-\alpha} + \sum_{i=1}^{n}(i + c)^{-\alpha} \le \frac{1}{1 - \alpha}(n + c)^{1-\alpha}.$$

*Proof.* As $(u + c)^{-\alpha}$ is monotonically decreasing, bounding by integration,

$$\sum_{i=1}^{n}(i + c)^{-\alpha} \le \sum_{i=1}^{n} \int_{i-1}^{i}(u + c)^{-\alpha}du = \int_0^n (u + c)^{-\alpha}du = \frac{1}{1 - \alpha}\left((n + c)^{1-\alpha} - c^{1-\alpha}\right).$$

We conclude by noting that for $c \ge 1$ and $\alpha \in (0, 1)$, $c^{1-\alpha}/(1 - \alpha) \ge c^{-\alpha} \ge (1 + c)^{-\alpha}$. $\qquad \square$

**Lemma 5.** *Let $x \geq 1$ and $\alpha \in (0, 1)$. Then*

$$x^{-\alpha} - (x+1)^{-\alpha} \leq \frac{\alpha}{x(x+1)^\alpha}.$$

*Proof.* The inequality follows directly from Bernoulli's inequality, $(1 + x)^r \leq 1 + rx$, which holds for $r \in [0, 1]$ and $x \geq -1$. Using the inequality,

$$x^{-\alpha} - (x+1)^{-\alpha} = (x+1)^{-\alpha}\left(\left(1 + \frac{1}{x}\right)^\alpha - 1\right) \leq \frac{\alpha}{x(x+1)^\alpha}. \qquad \square$$

## G  Proofs of Section 2

In this section we give formal proofs of Theorems 1 and 2.

### G.1  Regret analysis for stochastic convex smooth optimization

We first present a key lemma, which may be of independent interest. The lemma establish a regret bound, which is an improved version of Lemma 2.1 of Zamani and Glineur [2023], specialized for stochastic optimization with smooth objectives. The result holds in a general setting that accommodates both *with*-replacement and *without*-replacement sampling schemes, under a unified sampling assumption stated in the lemma.

**Lemma 6.** *Let $f : \mathbb{R}^d \times Z \to \mathbb{R}$ be such that $f(\cdot; z)$ is $\beta$-smooth and convex for every $z$. Further, let $z_1, \ldots, z_T \sim \mathcal{Z}(T)$ where $\mathcal{Z}(T)$ is a distribution over $Z^T$ that satisfies the following: for any $s \leq t$, conditioned on $z_1, \ldots, z_{s-1}$, it holds that $z_s$ and $z_t$ are identically distributed. Then for any $x^\star \in \mathbb{R}^d$ and any weight sequence $0 < v_0 \leq v_1 \leq \cdots \leq v_T$, running SGD with initialization $x_1 \in \mathbb{R}^d$ and stepsize sequence $\eta_1, \ldots, \eta_T > 0$,*

$$t = 1, \ldots, T : \quad x_{t+1} = x_t - \eta_t \nabla f_t(x_t), \quad \text{where } f_t := f(\cdot; z_t),$$

*it holds that*

$$\sum_{t=1}^T c_t \mathbb{E}[f_t(x_t) - f_t(x^\star)] \leq \frac{1}{2} v_0^2 \|x_1 - x^\star\|^2 + \frac{1}{2} \sum_{t=1}^T \eta_t^2 v_t^2 \mathbb{E}\|\nabla f_t(x_t)\|^2$$

$$- \frac{1}{2\beta} \sum_{t=1}^T \eta_t v_t \left(\sum_{s=1}^t (v_s - v_{s-1}) \mathbb{E}\|\nabla f_t(x_t) - \nabla f_t(x_s)\|^2 + v_0 \mathbb{E}\|\nabla f_t(x_t) - \nabla f_t(x^\star)\|^2\right),$$

*where $c_t := \eta_t v_t^2 - (v_t - v_{t-1}) \sum_{s=t}^T \eta_s v_s$.*

*Proof.* Define $y_1, \ldots, y_T$ recursively by $y_0 = x^\star$ and for $t \geq 1$:

$$y_t = \frac{v_{t-1}}{v_t} y_{t-1} + \left(1 - \frac{v_{t-1}}{v_t}\right) x_t.$$

Observe that:

$$\|x_{t+1} - y_{t+1}\|^2 = \frac{v_t^2}{v_{t+1}^2} \|x_{t+1} - y_t\|^2 = \frac{v_t^2}{v_{t+1}^2} \|x_t - \eta_t \nabla f_t(x_t) - y_t\|^2$$

$$= \frac{v_t^2}{v_{t+1}^2} \left(\|x_t - y_t\|^2 - 2\eta_t \langle \nabla f_t(x_t), x_t - y_t \rangle + \eta_t^2 \|\nabla f_t(x_t)\|^2\right).$$

Thus, by rearranging, we obtain

$$\eta_t v_t^2 \langle \nabla f_t(x_t), x_t - y_t \rangle = \frac{1}{2} v_t^2 \|x_t - y_t\|^2 - \frac{1}{2} v_{t+1}^2 \|x_{t+1} - y_{t+1}\|^2 + \frac{1}{2} \eta_t^2 v_t^2 \|\nabla f_t(x_t)\|^2.$$

Summing over $t = 1, \ldots, T$ yields

$$\sum_{t=1}^T \eta_t v_t^2 \langle \nabla f_t(x_t), x_t - y_t \rangle \leq \frac{1}{2} v_0^2 \|x_1 - x^\star\|^2 + \frac{1}{2} \sum_{t=1}^T \eta_t^2 v_t^2 \|\nabla f_t(x_t)\|^2, \qquad (9)$$

where we used that

$$\|x_1 - y_1\| = \frac{v_0}{v_1}\|x_1 - y_0\| = \frac{v_0}{v_1}\|x_1 - x^\star\|.$$

On the other hand, $y_t$ can be written directly as a convex combination of $x_1, \ldots, x_T$ and $x^\star$, as follows:

$$y_t = \frac{v_0}{v_t}x^\star + \sum_{s=1}^{t}\frac{v_s - v_{s-1}}{v_t}x_s.$$

Using a standard property of convex and smooth functions [e.g., Nesterov, 1998], for any $x \in \mathbb{R}^d$,

$$\langle \nabla f_t(x_t), x_t - x \rangle \geq f_t(x_t) - f_t(x) + \frac{1}{2\beta}\|\nabla f_t(x_t) - \nabla f_t(x)\|^2.$$

Hence, as $v_s \geq v_{s-1}$,

$$\langle \nabla f_t(x_t), x_t - y_t \rangle = \sum_{s=1}^{t}\frac{v_s - v_{s-1}}{v_t}\langle \nabla f_t(x_t), x_t - x_s \rangle + \frac{v_0}{v_t}\langle \nabla f_t(x_t), x_t - x^\star \rangle$$

$$\geq \sum_{s=1}^{t}\frac{v_s - v_{s-1}}{v_t}\left(f_t(x_t) - f_t(x_s) + \frac{1}{2\beta}\|\nabla f_t(x_t) - \nabla f_t(x_s)\|^2\right)$$

$$+ \frac{v_0}{v_t}\left(f_t(x_t) - f_t(x^\star) + \frac{1}{2\beta}\|\nabla f_t(x_t) - \nabla f_t(x^\star)\|^2\right)$$

$$= f_t(x_t) - \sum_{s=1}^{t}\frac{v_s - v_{s-1}}{v_t}f_t(x_s) - \frac{v_0}{v_t}f_t(x^\star) + \frac{1}{2\beta}\sum_{s=1}^{t}\frac{v_s - v_{s-1}}{v_t}\|\nabla f_t(x_t) - \nabla f_t(x_s)\|^2$$

$$+ \frac{v_0}{2\beta v_t}\|\nabla f_t(x_t) - \nabla f_t(x^\star)\|^2.$$

Multiplying by $\eta_t v_t^2$ and plugging back to Eq. (9),

$$\frac{1}{2}v_0^2\|x_1 - x^\star\|^2 + \frac{1}{2}\sum_{t=1}^{T}\eta_t^2 v_t^2\|\nabla f_t(x_t)\|^2 \geq \sum_{t=1}^{T}\eta_t v_t\left(v_t f_t(x_t) - \sum_{s=1}^{t}(v_s - v_{s-1})f_t(x_s) - v_0 f_t(x^\star)\right)$$

$$+ \frac{1}{2\beta}\sum_{t=1}^{T}\eta_t v_t\left(\sum_{s=1}^{t}(v_s - v_{s-1})\|\nabla f_t(x_t) - \nabla f_t(x_s)\|^2 + v_0\|\nabla f_t(x_t) - \nabla f_t(x^\star)\|^2\right)$$

$$= \sum_{t=1}^{T}\eta_t v_t\left(v_t(f_t(x_t) - f_t(x^\star)) - \sum_{s=1}^{t}(v_s - v_{s-1})(f_t(x_s) - f_t(x^\star))\right)$$

$$+ \frac{1}{2\beta}\sum_{t=1}^{T}\eta_t v_t\left(\sum_{s=1}^{t}(v_s - v_{s-1})\|\nabla f_t(x_t) - \nabla f_t(x_s)\|^2 + v_0\|\nabla f_t(x_t) - \nabla f_t(x^\star)\|^2\right).$$

Taking expectation, noting that for all $t \geq s$, $\mathbb{E}f_t(x_s) - f_t(x^\star) = \mathbb{E}f_s(x_s) - f_s(x^\star)$ (since $f_t, f_s$ are identically distributed condition on $x_s$ and $x^\star$ is fixed), and rearranging the terms,

$$\frac{1}{2}v_0^2\|x_1 - x^\star\|^2 + \frac{1}{2}\sum_{t=1}^{T}\eta_t^2 v_t^2\mathbb{E}\|\nabla f_t(x_t)\|^2 \geq \sum_{t=1}^{T}\mathbb{E}[f_t(x_t) - f_t(x^\star)]\left(\eta_t v_t^2 - (v_t - v_{t-1})\sum_{s=t}^{T}\eta_s v_s\right)$$

$$+ \frac{1}{2\beta}\sum_{t=1}^{T}\eta_t v_t\left(\sum_{s=1}^{t}(v_s - v_{s-1})\mathbb{E}\|\nabla f_t(x_t) - \nabla f_t(x_s)\|^2 + v_0\mathbb{E}\|\nabla f_t(x_t) - \nabla f_t(x^\star)\|^2\right).$$

We conclude by rearranging the terms and plugging $c_t$. $\qquad\square$

## G.2 Proof of Theorem 1

We now give the formal proof of Theorem 1. In addition to Lemma 6, we will use the following lemma, which handles the cross terms $\mathbb{E}\|\nabla f_t(x_t) - \nabla f_t(x_s)\|^2$ which are introduced by the regret bound of Lemma 6. See Appendix G.4 for the proof.

**Lemma 7.** *Let $f: \mathbb{R}^d \times Z \to \mathbb{R}$ be such that $f(\cdot; z)$ is $\beta$-smooth and convex for every $z$. Further, let $z_1, \ldots, z_T \sim \mathcal{Z}(T)$ where $\mathcal{Z}(T)$ is a distribution over $Z^T$ that satisfies the following: for any $s \leq t$, conditioned on $z_1, \ldots, z_{s-1}$, it holds that $z_s$ and $z_t$ are identically distributed. Let $\alpha \in (0, \frac{1}{2})$, let $v_t = (T - t + 2)^{-\alpha}$ for any $t \in [T-1]$, and let $v_T = v_{T-1}$. Then for any $x^\star \in \mathbb{R}^d$, running SGD with initialization $x_1 \in \mathbb{R}^d$ and stepsize sequence $\eta_1, \ldots, \eta_T > 0$,*

$$t = 1, \ldots, T: \quad x_{t+1} = x_t - \eta_t \nabla f_t(x_t), \quad \text{where } f_t := f(\cdot; z_t),$$

*it holds that*

$$\sum_{t=1}^T v_t \left( \sum_{s=1}^t (v_s - v_{s-1}) \mathbb{E} \|\nabla f_t(x_t) - \nabla f_t(x_s)\|^2 + v_0 \mathbb{E} \|\nabla f_t(x_t) - \nabla f_t(x^\star)\|^2 \right)$$

$$\geq (1 - 2\alpha) v_T^2 \mathbb{E} \|\nabla f_T(x_T) - \nabla f_T(x^\star)\|^2 + \sum_{t=1}^{T-1} \left( (1 - 4\alpha) v_t^2 + (v_t - v_{t-1}) \sum_{s=t}^T v_s \right) \mathbb{E} \|\nabla f_t(x_t) - \nabla f_t(x^\star)\|^2.$$

We proceed to prove Theorem 1.

*Proof of Theorem 1.* Let $\alpha \in (0, \frac{1}{2})$, $v_t = (T - t + 2)^{-\alpha}$ for $t \in [T-1]$, $v_T = v_{T-1}$, and denote $\tilde{f}_t(x) := f_t(x) - \langle \nabla f_t(x^\star), x \rangle$ such that $\nabla \tilde{f}_t(x) = \nabla f_t(x) - \nabla f_t(x^\star)$. By Lemma 6 with fixed stepsize $\eta_t = \eta$ and weights $0 < v_1 \leq \cdots \leq v_T$, which is applicable under with-replacement sampling,

$$\frac{1}{2} v_0^2 \|x_1 - x^\star\|^2 + \frac{1}{2} \sum_{t=1}^T \eta^2 v_t^2 \mathbb{E} \|\nabla f_t(x_t)\|^2 \geq \sum_{t=1}^T \mathbb{E}[f_t(x_t) - f_t(x^\star)] \left( \eta v_t^2 - (v_t - v_{t-1}) \sum_{s=t}^T \eta v_s \right)$$

$$+ \frac{1}{2\beta} \sum_{t=1}^T \eta v_t \left( \sum_{s=1}^t (v_s - v_{s-1}) \mathbb{E} \|\nabla \tilde{f}_t(x_t) - \nabla \tilde{f}_t(x_s)\|^2 + v_0 \mathbb{E} \|\nabla \tilde{f}_t(x_t)\|^2 \right),$$

where we substituted $\nabla f_t(x_t) - \nabla f_t(x_s) = \nabla \tilde{f}_t(x_t) - \nabla \tilde{f}_t(x_s)$. Dividing by $\eta$, plugging $v_T = v_{T-1}$, and denoting $a_t = (v_t - v_{t-1}) \sum_{s=t}^T v_s$,

$$\frac{v_0^2}{2\eta} \|x_1 - x^\star\|^2 + \frac{\eta}{2} \sum_{t=1}^T v_t^2 \mathbb{E} \|\nabla f_t(x_t)\|^2 \geq v_T^2 \mathbb{E}[f_T(x_T) - f_T(x^\star)] + \sum_{t=1}^{T-1} \mathbb{E}[f_t(x_t) - f_t(x^\star)](v_t^2 - a_t)$$

$$+ \frac{1}{2\beta} \sum_{t=1}^T v_t \left( \sum_{s=1}^t (v_s - v_{s-1}) \mathbb{E} \|\nabla \tilde{f}_t(x_t) - \nabla \tilde{f}_t(x_s)\|^2 + v_0 \mathbb{E} \|\nabla \tilde{f}_t(x_t)\|^2 \right).$$

The technical Lemmas 4 and 5, which are proved in Appendix F, state that for any $c \geq 1$, $\alpha \in (0, 1)$, and $n \in \mathbb{N}$, it holds that $(1 + c)^{-\alpha} + \sum_{i=1}^n (i + c)^{-\alpha} \leq \frac{1}{1-\alpha}(n + c)^{1-\alpha}$, and that for any $x \geq 1$, and $\alpha \in (0, 1)$, it holds that $x^{-\alpha} - (x + 1)^{-\alpha} \leq \frac{\alpha}{x(x+1)^\alpha}$. Using these, for $t < T$,

$$a_t \leq \frac{\alpha}{(T - t + 2)(T - t + 3)^\alpha} \cdot \frac{1}{1 - \alpha} (T - t + 2)^{1-\alpha} \leq \frac{\alpha}{1 - \alpha} v_t^2 \leq v_t^2,$$

where we used the fact that $\alpha \in (0, \frac{1}{2})$. Hence, we can apply the standard inequality for convex smooth functions [e.g., Nesterov, 1998], $\|\nabla f_t(x_t) - \nabla f_t(x^\star)\|^2 \leq 2\beta(f_t(x_t) - \langle \nabla f_t(x^\star), x_t - x^\star \rangle)$, and after taking expectation and substituting $\nabla \tilde{f}_t(x_t) = \nabla f_t(x_t) - \nabla f_t(x^\star)$, for any $t \in [T]$,

$$\mathbb{E} \|\nabla \tilde{f}_t(x_t)\|^2 \leq 2\beta \mathbb{E}[(f_t(x_t) - f_t(x^\star) - \langle \nabla F(x^\star), x_t - x^\star \rangle)] = 2\beta \mathbb{E}[f_t(x_t) - f_t(x^\star)], \tag{10}$$

where we used that $f_t$ is independent of $x_t$ and $x^\star$ is a minimizer of $F(x)$, obtaining that

$$\frac{v_0^2}{2\eta} \|x_1 - x^\star\|^2 + \frac{\eta}{2} \sum_{t=1}^T v_t^2 \mathbb{E} \|\nabla f_t(x_t)\|^2 \geq v_T^2 \mathbb{E}[f_T(x_T) - f_T(x^\star)] + \frac{1}{2\beta} \sum_{t=1}^{T-1} \mathbb{E} \|\nabla \tilde{f}_t(x_t)\|^2 (v_t^2 - a_t)$$

$$+ \frac{1}{2\beta} \sum_{t=1}^T v_t \left( \sum_{s=1}^t (v_s - v_{s-1}) \mathbb{E} \|\nabla \tilde{f}_t(x_t) - \nabla \tilde{f}_t(x_s)\|^2 + v_0 \mathbb{E} \|\nabla \tilde{f}_t(x_t)\|^2 \right).$$

Plugging Lemma 7 (and substituting $\nabla f_t(x) - \nabla f_t(x^\star) = \nabla \tilde{f}_t(x_t)$),

$$\frac{v_0^2}{2\eta}\|x_1 - x^\star\|^2 + \frac{\eta}{2}\sum_{t=1}^{T} v_t^2 \mathbb{E}\|\nabla f_t(x_t)\|^2 \geq v_T^2 \mathbb{E}[f_T(x_T) - f_T(x^\star)] + \frac{1}{2\beta}\sum_{t=1}^{T-1} \mathbb{E}\|\nabla \tilde{f}_t(x_t)\|^2 (v_t^2 - a_t)$$

$$+ \frac{1}{2\beta}\left((1 - 2\alpha)v_T^2 \mathbb{E}\|\nabla \tilde{f}_T(x_T)\|^2 + \sum_{t=1}^{T-1}\left((1 - 4\alpha)v_t^2 + (v_t - v_{t-1})\sum_{s=t}^{T} v_s\right)\mathbb{E}\|\nabla \tilde{f}_t(x_t)\|^2\right).$$

Rearranging the terms and using the definition of $a_t$,

$$v_T^2 \mathbb{E}[f_T(x_T) - f_T(x^\star)] \leq \frac{v_0^2}{2\eta}\|x_1 - x^\star\|^2 + \frac{\eta}{2}\sum_{t=1}^{T} v_t^2 \mathbb{E}\|\nabla f_t(x_t)\|^2 - \frac{1}{\beta}\sum_{t=1}^{T-1}(1 - 2\alpha)v_t^2 \mathbb{E}\|\nabla \tilde{f}_t(x_t)\|^2$$

$$- \frac{v_T^2}{2\beta}(1 - 2\alpha)\mathbb{E}\|\nabla \tilde{f}_T(x_T)\|^2.$$

By Eq. (10), $-(1 - 2\alpha)\mathbb{E}[f_T(x_T) - f_T(x^\star)] \leq -\frac{1-2\alpha}{2\beta}\mathbb{E}\|\nabla \tilde{f}_T(x_T)\|^2$. Adding to the above display,

$$2\alpha v_T^2 \mathbb{E}[f_T(x_T) - f_T(x^\star)] \leq \frac{v_0^2}{2\eta}\|x_1 - x^\star\|^2 + \frac{\eta}{2}\sum_{t=1}^{T} v_t^2 \mathbb{E}\|\nabla f_t(x_t)\|^2$$

$$- \frac{1}{\beta}\sum_{t=1}^{T}(1 - 2\alpha)v_t^2 \mathbb{E}\|\nabla \tilde{f}_t(x_t)\|^2. \tag{11}$$

As $x^\star \in \arg\min_{x \in \mathbb{R}^d} f(x; z)$ for every $z \in Z$, $\nabla \tilde{f}_t(x_t) = \nabla f(x_t)$, and by setting $\alpha = \frac{2-\beta\eta}{4} \in (0, \frac{1}{2})$,

$$\frac{2 - \beta\eta}{2} v_T^2 \mathbb{E}[f_T(x_T) - f_T(x^\star)] \leq \frac{v_0^2}{2\eta}\|x_1 - x^\star\|^2.$$

Substituting $v_0 = (T + 2)^{-\alpha}$ and $v_T = 3^{-\alpha}$ and rearranging,

$$\mathbb{E}[f_T(x_T) - f_T(x^\star)] \leq \frac{3^{2\alpha}\|x_1 - x^\star\|^2}{\eta(2 - \beta\eta)(T + 2)^{2\alpha}}.$$

Hence, substituting $\alpha = \frac{2-\beta\eta}{4}$ and since $2\alpha \leq 1$,

$$\mathbb{E}[f_T(x_T) - f_T(x^\star)] \leq \frac{3\|x_1 - x^\star\|^2}{\eta(2 - \beta\eta)(T + 2)^{1-\beta\eta/2}} \leq \frac{3\|x_1 - x^\star\|^2}{\eta(2 - \beta\eta)T^{1-\beta\eta/2}}. \tag{12}$$

We conclude by noting that $\mathbb{E}[f_T(x_T) - f_T(x^\star)] = \mathbb{E}[F(x_T) - F(x^\star)]$. $\qquad\square$

### G.3   Proof of Theorem 2

Note that Lemmas 6 and 7 used to prove Theorem 1 hold without the assumption that $x^\star$ is the minimizer of each function. Hence, the argument in the proof of Theorem 1 up to the point of Eq. (11) is applicable in the setting of Theorem 2, showing that

$$2\alpha v_T^2 \mathbb{E}[f_T(x_T) - f_T(x^\star)] \leq \frac{v_0^2}{2\eta}\|x_1 - x^\star\|^2 + \frac{\eta}{2}\sum_{t=1}^{T} v_t^2 \mathbb{E}\|\nabla f_t(x_t)\|^2$$

$$- \frac{1}{\beta}\sum_{t=1}^{T}(1 - 2\alpha)v_t^2 \mathbb{E}\|\nabla \tilde{f}_t(x_t)\|^2. \tag{13}$$

Let $\epsilon = 1 - 2\alpha - \frac{\beta\eta}{2} > 0$, restricting $\alpha < (2 - \beta\eta)/4$. By Young's inequality,

$$\mathbb{E}\|\nabla f_t(x_t)\|^2 = \mathbb{E}\left[\|\nabla \tilde{f}_t(x_t)\|^2 + 2\langle\nabla \tilde{f}_t(x_t), \nabla f_t(x^\star)\rangle + \|\nabla f_t(x^\star)\|^2\right]$$

$$\leq \left(1 + \frac{2\epsilon}{\beta\eta}\right)\mathbb{E}\|\nabla \tilde{f}_t(x_t)\|^2 + \left(1 + \frac{\beta\eta}{2\epsilon}\right)\mathbb{E}\|\nabla f_t(x^\star)\|^2.$$

Thus, as $\epsilon = 1 - 2\alpha - \frac{\beta\eta}{2}$, for any $t \in [T]$,

$$\frac{\eta}{4}v_t^2 \mathbb{E}\|\nabla f_t(x_t)\|^2 \leq \frac{\eta}{4}v_t^2\left(\left(1 + \frac{2\epsilon}{\beta\eta}\right)\mathbb{E}\|\nabla \tilde{f}_t(x_t)\|^2 + \left(1 + \frac{\beta\eta}{2\epsilon}\right)\mathbb{E}\|\nabla f_t(x^\star)\|^2\right)$$

$$= \frac{\eta}{4}v_t^2\frac{2 - 4\alpha}{\beta\eta}\mathbb{E}\|\nabla \tilde{f}_t(x_t)\|^2 + \frac{\eta}{4}\left(1 + \frac{\beta\eta}{2\epsilon}\right)v_t^2\mathbb{E}\|\nabla f_t(x^\star)\|^2$$

$$= \frac{1}{2\beta}v_t^2(1 - 2\alpha)\mathbb{E}\|\nabla \tilde{f}_t(x_t)\|^2 + \frac{\eta}{4}\left(1 + \frac{\beta\eta}{2\epsilon}\right)v_t^2\mathbb{E}\|\nabla f_t(x^\star)\|^2.$$

Plugging back to Eq. (13) and substituting $\mathbb{E}[f_T(x_T) - f_T(x^\star)] = \mathbb{E}[f(x_T) - f(x^\star)]$,

$$\alpha v_T^2 \mathbb{E}[F(x_T) - F(x^\star)] \leq \frac{v_0^2}{4\eta}\|x_1 - x^\star\|^2 + \frac{\eta}{4}\left(1 + \frac{\beta\eta}{2\epsilon}\right)\sum_{t=1}^{T} v_t^2\mathbb{E}\|\nabla f_t(x^\star)\|^2$$

$$= \frac{v_0^2}{4\eta}\|x_1 - x^\star\|^2 + \frac{\eta}{4}\left(1 + \frac{\beta\eta}{2\epsilon}\right)\sigma_\star^2\sum_{t=1}^{T} v_t^2,$$

where the last equality follows from $\mathbb{E}\|\nabla f_t(x^\star)\|^2 = \mathbb{E}_{z\sim\mathcal{Z}}\|\nabla f(x^\star; z)\|^2 = \sigma_\star^2$. Lemma 4, which is proved in Appendix F, state that for any $c \geq 1$, $\alpha \in (0, 1)$, and $n \in \mathbb{N}$, it holds that $(1 + c)^{-\alpha} + \sum_{i=1}^{n}(i + c)^{-\alpha} \leq \frac{1}{1-\alpha}(n + c)^{1-\alpha}$. Hence,

$$\sum_{t=1}^{T} v_t^2 \leq \frac{1}{1 - 2\alpha}(T + 1)^{1-2\alpha} \leq \frac{1}{1 - 2\alpha}(T + 2)^{1-2\alpha}.$$

Thus,

$$\alpha v_T^2 \mathbb{E}[F(x_T) - F(x^\star)] \leq \frac{v_0^2}{4\eta}\|x_1 - x^\star\|^2 + \frac{\eta\left(1 + \frac{\beta\eta}{2\epsilon}\right)\sigma_\star^2(T + 2)^{1-2\alpha}}{4(1 - 2\alpha)}.$$

Substituting $v_T = 3^{-\alpha} \geq 1/\sqrt{3}$ (since $\alpha < \frac{1}{2}$), $v_0 = (T + 2)^{-\alpha}$, and $\alpha = \frac{2-\beta\eta-2\epsilon}{4}$, and rearranging,

$$\mathbb{E}[F(x_T) - F(x^\star)] \leq \frac{3\|x_1 - x^\star\|^2}{(2 - \beta\eta - 2\epsilon)\eta(T + 2)^{1-\frac{\beta\eta}{2}-\epsilon}} + \frac{3\eta\sigma_\star^2(T + 2)^{\frac{\beta\eta}{2}+\epsilon}}{(2 - \beta\eta - 2\epsilon)\epsilon}.$$

Now set $\alpha = \frac{2-\beta\eta}{4} - \frac{2-\beta\eta}{4\log_2(T+2)} \in (0, \frac{1}{2})$, for which $\epsilon = \frac{2-\beta\eta}{2\log_2(T+2)}$, and note that $(T + 2)^\epsilon \leq (T + 2)^{1/\log_2(T+2)} \leq 2$ and $(2 - \beta\eta - 2\epsilon) \geq \frac{2-\beta\eta}{2}$ as $T \geq 2$. Thus,

$$\mathbb{E}[F(x_T) - F(x^\star)] \leq \frac{12\|x_1 - x^\star\|^2}{\eta(2 - \beta\eta)(T + 2)^{1-\frac{\beta\eta}{2}}} + \frac{24\eta\sigma_\star^2(T + 2)^{\frac{\beta\eta}{2}}\log_2(T + 2)}{(2 - \beta\eta)^2}$$

$$\leq \frac{12\|x_1 - x^\star\|^2}{\eta(2 - \beta\eta)T^{1-\frac{\beta\eta}{2}}} + \frac{48\eta\sigma_\star^2 T^{\frac{\beta\eta}{2}}\log_2(T + 2)}{(2 - \beta\eta)^2},$$

where that last inequality follows by $T \geq 2$ and $\beta\eta < 2$. The second claim, for when

$$\eta = \min\left\{\frac{1}{\beta\log_2(T)}, \frac{\|x_1 - x^\star\|}{\sigma_\star\sqrt{T\log_2(T + 2)}}\right\} \leq \frac{1}{\beta}, \qquad \text{(as } \log_2(T) \geq 1 \text{ since } T \geq 2\text{)}$$

follows by noting that $T^{\beta\eta/2} \leq T^{1/2\log_2(T)} = \sqrt{2} < 2$, so that

$$\mathbb{E}[F(x_T) - F(x^\star)] \leq \frac{12\|x_1 - x^\star\|^2}{\eta T^{1-\frac{\beta\eta}{2}}} + 48\eta\sigma_\star^2 T^{\frac{\beta\eta}{2}}\log_2(T + 2)$$

$$\leq \frac{24\beta\|x_1 - x^\star\|^2\log_2(T)}{T} + \frac{120\sigma_\star\|x_1 - x^\star\|\sqrt{\log_2(T + 2)}}{\sqrt{T}}. \qquad \square$$

## G.4 Proof of Lemma 7

Let $\tilde{f}_t(x) := f_t(x) - \langle \nabla f_t(x^\star), x \rangle$ and note that $\nabla \tilde{f}_t(x) = \nabla f_t(x) - \nabla f_t(x^\star)$. Hence, as

$$\|\nabla \tilde{f}_t(x_t) - \nabla \tilde{f}_t(x_s)\|^2 = \|\nabla \tilde{f}_t(x_t)\|^2 + \|\nabla \tilde{f}_t(x_s)\|^2 - 2\langle \nabla \tilde{f}_t(x_t), \nabla \tilde{f}_t(x_s) \rangle,$$

it holds that

$$\sum_{t=1}^{T} v_t \left( \sum_{s=1}^{t} (v_s - v_{s-1}) \mathbb{E}\|\nabla \tilde{f}_t(x_t) - \nabla \tilde{f}_t(x_s)\|^2 + v_0 \mathbb{E}\|\nabla \tilde{f}_t(x_t)\|^2 \right) \tag{14}$$

$$= \sum_{t=1}^{T} v_t \left( v_t \mathbb{E}\|\nabla \tilde{f}_t(x_t)\|^2 + \sum_{s=1}^{t} (v_s - v_{s-1}) \left( \mathbb{E}\|\nabla \tilde{f}_t(x_s)\|^2 - 2\mathbb{E}\langle \nabla \tilde{f}_t(x_t), \nabla \tilde{f}_t(x_s) \rangle \right) \right).$$

Next, we will focus on the summations of $\mathbb{E}\langle \nabla \tilde{f}_t(x_t), \nabla \tilde{f}_t(x_s) \rangle$. Rearranging,

$$\sum_{t=1}^{T} \sum_{s=1}^{t} v_t (v_s - v_{s-1}) \mathbb{E}\langle \nabla \tilde{f}_t(x_t), \nabla \tilde{f}_t(x_s) \rangle = \sum_{s=1}^{T} \sum_{t=s}^{T} v_t (v_s - v_{s-1}) \mathbb{E}\langle \nabla \tilde{f}_t(x_t), \nabla \tilde{f}_t(x_s) \rangle$$

$$= \sum_{s=1}^{T-1} \sum_{t=s}^{T} v_t (v_s - v_{s-1}) \mathbb{E}\langle \nabla \tilde{f}_t(x_t), \nabla \tilde{f}_t(x_s) \rangle,$$

where the last inequality follows by $v_T = v_{T-1}$. For $t \in [T]$ and $s \in [T-1]$, let

$$\lambda_{t,s} = c \cdot \frac{((T-s+1)^{-0.5} - (T-s+2)^{-0.5})}{(T-t+1)^{-0.5}} \cdot \frac{v_t}{v_s - v_{s-1}},$$

where $c > 0$ will be determined later. Noting that $v_t \geq v_{t-1}$ and using Young's inequality,

$$\sum_{t=1}^{T} \sum_{s=1}^{t} v_t (v_s - v_{s-1}) \mathbb{E}\langle \nabla \tilde{f}_t(x_t), \nabla \tilde{f}_t(x_s) \rangle \leq \frac{1}{2} \sum_{s=1}^{T-1} \sum_{t=s}^{T} v_t (v_s - v_{s-1}) \left( \lambda_{t,s} \tilde{G}_t^2 + \frac{\tilde{G}_s^2}{\lambda_{t,s}} \right), \tag{15}$$

where $\tilde{G}_t^2 := \mathbb{E}\|\nabla \tilde{f}_t(x_t)\|^2$, such that $\tilde{G}_s^2 = \mathbb{E}\|\nabla \tilde{f}_s(x_s)\|^2 = \mathbb{E}\|\nabla \tilde{f}_t(x_s)\|^2$ as for $s \leq t$, $f_t$ and $f_s$ are identically distributed conditioned on $x_s$. Denoting $\lambda_{t,T} = 0$,

$$\sum_{s=1}^{T-1} \sum_{t=s}^{T} v_t (v_s - v_{s-1}) \lambda_{t,s} \tilde{G}_t^2 = \sum_{s=1}^{T} \sum_{t=s}^{T} v_t (v_s - v_{s-1}) \lambda_{t,s} \tilde{G}_t^2 = \sum_{t=1}^{T} \sum_{s=1}^{t} v_t (v_s - v_{s-1}) \lambda_{t,s} \tilde{G}_t^2$$

$$\leq c \sum_{t=1}^{T} \tilde{G}_t^2 v_t^2 (T-t+1)^{0.5} \sum_{s=1}^{t} ((T-s+1)^{-0.5} - (T-s+2)^{-0.5})$$

$$\leq c \sum_{t=1}^{T} \tilde{G}_t^2 v_t^2, \tag{16}$$

and

$$\sum_{s=1}^{T-1} \sum_{t=s}^{T} v_t (v_s - v_{s-1}) \frac{\tilde{G}_s^2}{\lambda_{t,s}} = \frac{1}{c} \sum_{s=1}^{T-1} \frac{(v_s - v_{s-1})^2}{(T-s+1)^{-0.5} - (T-s+2)^{-0.5}} \tilde{G}_s^2 \sum_{t=s}^{T} (T-t+1)^{-0.5}$$

$$\leq \frac{2}{c} \sum_{s=1}^{T-1} \frac{(v_s - v_{s-1})^2}{(T-s+1)^{-0.5} - (T-s+2)^{-0.5}} \tilde{G}_s^2 (T-s+1)^{0.5}$$

$$= \frac{2}{c} \sum_{s=1}^{T-1} \tilde{G}_s^2 (v_s - v_{s-1})^2 \frac{(T-s+1)(T-s+2)^{0.5}}{(T-s+2)^{0.5} - (T-s+1)^{0.5}}$$

$$= \frac{2}{c} \sum_{s=1}^{T-1} \tilde{G}_s^2 (v_s - v_{s-1})^2 (T-s+1)(T-s+2)^{0.5}((T-s+2)^{0.5} + (T-s+1)^{0.5})$$

$$\leq \frac{4}{c} \sum_{s=1}^{T-1} \tilde{G}_s^2 (v_s - v_{s-1})^2 (T-s+2)^2,$$

where we bounded

$$\sum_{t=s}^{T}(T-t+1)^{-0.5} \le 1 + \sum_{t=s}^{T-1}\int_{t}^{t+1}(T-u+1)^{-0.5}du = 1 + \int_{s}^{T}(T-u+1)^{-0.5}du$$

$$\le 2(T-s+1)^{0.5}.$$

Note that by Lemma 5,

$$(v_s - v_{s-1})^2 \le \left(\frac{\alpha}{(T-s+2)(T-s+3)^{\alpha}}\right)^2 = \frac{\alpha^2 v_{s-1}^2}{(T-s+2)^2}.$$

Thus,

$$\sum_{t=1}^{T-1}\sum_{s=1}^{t} v_t(v_s - v_{s-1})\frac{\tilde{G}_s^2}{\lambda_{t,s}} \le \frac{4\alpha^2}{c}\sum_{s=1}^{T-1}\tilde{G}_s^2 v_{s-1}^2 \le \frac{4\alpha^2}{c}\sum_{s=1}^{T-1}\tilde{G}_s^2 v_s^2. \qquad (v_s \ge v_{s-1})$$

Together with Eq. (16), plugging to Eq. (15) and setting $c = 2\alpha$,

$$\sum_{t=1}^{T}\sum_{s=1}^{t} v_t(v_s - v_{s-1})\mathbb{E}\langle\nabla\tilde{f}_t(x_t), \nabla\tilde{f}_t(x_s)\rangle \le \alpha v_T^2\tilde{G}_T^2 + \sum_{t=1}^{T-1}2\alpha v_t^2\tilde{G}_t^2.$$

Returning to Eq. (14),

$$\sum_{t=1}^{T} v_t\left(\sum_{s=1}^{t}(v_s - v_{s-1})\mathbb{E}\|\nabla\tilde{f}_t(x_t) - \nabla\tilde{f}_t(x_s)\|^2 + v_0\mathbb{E}\|\nabla\tilde{f}_t(x_t)\|^2\right)$$

$$\ge \sum_{t=1}^{T} v_t\left(v_t\tilde{G}_t^2 + \sum_{s=1}^{t}(v_s - v_{s-1})\tilde{G}_s^2\right) - 2\alpha v_T^2\tilde{G}_T^2 - \sum_{t=1}^{T-1}4\alpha v_t^2\tilde{G}_t^2,$$

where we used the fact that $\mathbb{E}\|\nabla\tilde{f}_t(x_s)\|^2 = \mathbb{E}\|\nabla\tilde{f}_s(x_s)\|^2 = \tilde{G}_s^2$ for $s \le t$ as $f_t$ and $f_s$ are identically distributed conditioned on $x_s$ (which implies that $\tilde{f}_t$, and $\tilde{f}_s$ are identically distributed conditioned in $x_s$). Rearranging the terms,

$$\sum_{t=1}^{T} v_t\left(\sum_{s=1}^{t}(v_s - v_{s-1})\mathbb{E}\|\nabla\tilde{f}_t(x_t) - \nabla\tilde{f}_t(x_s)\|^2 + v_0\mathbb{E}\|\nabla\tilde{f}_t(x_t)\|^2\right)$$

$$\ge (1 - 2\alpha)v_T^2\tilde{G}_T^2 + \sum_{t=1}^{T-1}\left((1 - 4\alpha)v_t^2 + (v_t - v_{t-1})\sum_{s=t}^{T}v_s\right)\tilde{G}_t^2,$$

where again we used the fact that $v_T = v_{T-1}$. We conclude by substituting $\tilde{G}_t^2 = \mathbb{E}\|\nabla\tilde{f}_t(x_t)\|^2$ and $\nabla\tilde{f}_t(x) = \nabla f_t(x) - \nabla f_t(x^{\star})$. $\qquad\square$

## H   Proofs of Section 3.1

### H.1   Proof of Theorem 3

The proof is based on the following two lemmas. The first is a convergence bound for $\mathbb{E}[f_T(x_T) - f_T(x^{\star})]$, which is an extension of Theorem 1 to SGD with sampling without replacement. Note that $\mathbb{E}f_T(x_T)$ is not equal to $\mathbb{E}f(x_T)$ (in general) when sampling without replacement due to the correlations between $f_T$ and $x_T$.

**Lemma 8.** *Let $f: \mathbb{R}^d \times Z \to \mathbb{R}$ be such that $f(\cdot; z)$ is $\beta$-smooth and convex for every $z$. Assume $\mathcal{Z}(T)$ is a distribution over $Z^T$ that satisfies the following: For $z_1, \ldots, z_T \sim \mathcal{Z}(T)$, conditioned on $z_1, \ldots, z_{s-1}$, for $s \le t$ it holds that $z_s$ and $z_t$ are identically distributed. Assume there exists a joint minimizer $x^{\star} \in \cap_{z\in Z}\arg\min_{x\in\mathbb{R}^d}f(x; z)$. Then, for SGD on $z_1, \ldots, z_T \sim \mathcal{Z}(T)$ initialized at $x_1 \in \mathbb{R}^d$ with step size $\eta < 2/\beta$;*

$$t = 1, \ldots, T: \quad x_{t+1} = x_t - \eta\nabla f_t(x_t), \quad where\ f_t := f(\cdot; z_t),$$

*where $T \geq 2$, we have the following last iterate guarantee:*

$$\mathbb{E}[f_T(x_T) - f_T(x^\star)] \leq \frac{3\|x_1 - x^\star\|^2}{\eta(2 - \beta\eta)T^{1-\frac{\beta\eta}{2}}}.$$

*In particular, when $\eta = \frac{1}{\beta}$, $\mathbb{E}[f_T(x_T) - f_T(x^\star)] \leq \frac{3\beta\|x_1 - x^\star\|^2}{\sqrt{T}}$.*

The second is a without-replacement generalization upper bound in the smooth realizable setting established in Lemma 34 of Evron et al. [2025].

**Lemma 9** (Evron et al. [2025]). *For without-replacement SGD Eq. (4) with step size $\eta \leq 2/\beta$, for all $2 \leq T \leq n$, we have that the following holds:*

$$\mathbb{E}_\pi \left[ \frac{1}{T-1} \sum_{t=1}^{T-1} f(x_T; \pi_t) - f(x^\star; \pi_t) \right] \leq 2\mathbb{E}_\pi[f(x_T; \pi_T) - f(x^\star; \pi_T)] + \frac{4\beta^2\eta\|x_1 - x^\star\|^2}{T}.$$

We note that the above statement is a minor adjustment of the original one, and follows from the last line of the proof given by Evron et al. [2025].

*Proof of Theorem 3.* By Lemma 9, we have for all $T \leq n$:

$$\mathbb{E}F_Z(x_T) - F_Z(x^\star)$$

$$= \frac{T-1}{n}\mathbb{E}\left[ \frac{1}{T-1} \sum_{t=1}^{T-1} f(x_T; \pi_t) - f(x^\star; \pi_t) \right] + \frac{n-T+1}{n}\mathbb{E}\left[ f(x_T; \pi_T) - f(x^\star; \pi_T) \right]$$

$$\leq \mathbb{E}\left[ \frac{1}{T-1} \sum_{t=1}^{T-1} f(x_T; \pi_t) - f(x^\star; \pi_t) \right] + \mathbb{E}\left[ f(x_T; \pi_T) - f(x^\star; \pi_T) \right]$$

$$\leq 3\mathbb{E}\left[ f(x_T; \pi_T) - f(x^\star; \pi_T) \right] + \frac{4\beta^2\eta\|x_1 - x^\star\|^2}{T}.$$

Now, since $\pi$ is a uniformly random permutation, $\pi_1, \ldots, \pi_T$ satisfy that conditioned on $\pi_1, \ldots, \pi_{s-1}$, we have that $\pi_s, \pi_t$ are identically distributed for $s \leq t$, as both are uniform over $Z \setminus \{z_1, \ldots z_{s-1}\}$. Hence Lemma 8 applies, and we have the that,

$$\mathbb{E}[f(x_T; \pi_T) - f_T(x^\star; \pi_T)] \leq \frac{3\|x_1 - x^\star\|^2}{\eta(2 - \beta\eta)(T + 2)^{1-\frac{\beta\eta}{2}}}.$$

Combining both upper bounds, the result follows. $\qquad\square$

## H.2 Proof of Lemma 8

The proof is identical to the proof of Theorem 1, beside the following points:

1. Lemmas 6 and 7 are applied under the general sampling scheme of Lemma 8, which is the same sampling scheme of Lemmas 6 and 7, instead of under the with-replacement sampling assumption of Theorem 1.

2. The bound $\mathbb{E}\|\nabla\tilde{f}_t(x_t)\|^2 \leq 2\beta\mathbb{E}[f_t(x_t) - f_t(x^\star)]$, which is established in Eq. (10) using the with-replacement assumption, is established instead using the realizability assumption (under which $\nabla f_t(x^\star) = 0$) by the following argument,

$$\mathbb{E}\|\nabla\tilde{f}_t(x_t)\|^2 = \mathbb{E}\|\nabla f_t(x_t)\|^2 \leq 2\beta\mathbb{E}[(f_t(x_t) - f_t(x^\star)] = 2\beta\mathbb{E}[f_t(x_t) - f_t(x^\star)].$$

As the rest of the argument does not exploit the with-replacement assumption up to (and including) Eq. (12), Eq. (12) holds and

$$\mathbb{E}[f_T(x_T) - f_T(x^\star)] \leq \frac{3\|x_1 - x^\star\|^2}{\eta(2 - \beta\eta)T^{1-\frac{\beta\eta}{2}}}.$$

The second inequality follows immediately by setting $\eta = \frac{1}{\beta}$. $\qquad\square$

# I Proofs of Sections 3.2 and 3.3

## I.1 Lemmas from Evron et al. [2025]

The proofs in this section use the following lemmas from Evron et al. [2025].

**Lemma 10** (Analogous to Reduction 2 in Evron et al. [2025]). *In the realizable case, under any ordering $\tau$, the block Kaczmarz method is equivalent to SGD with a step size of $\eta = c$, applied w.r.t. a convex, 1-smooth least squares objective: $\left\{ f_i(x) \triangleq \frac{1}{2} \|A_i^+(A_i x - b_i)\|^2 \right\}_{i=1}^m$. That is, the iterates $x_1, \ldots, x_T$ of Eq. (5) and SGD with step-size $\eta = c$ coincide.*

**Lemma 11** (Lemma 6 in Evron et al. [2025]). *Consider any realizable task collection such that $A_i x^\star = b_i$ for all $i \in [m]$. Define $f_i(x) = \frac{1}{2} \|A_i^+(A_i x - b_i)\|^2$. Then, $\forall i \in [m], x \in \mathbb{R}^d$*

(i) **Upper bound:** $\frac{1}{2} \|A_i x - b_i\|^2 \le R^2 f_i(x)$.

(ii) **Gradient:** $\quad \nabla_x f_i(x) = A_i^+ A_i w - A_i^+ b_i$.

(iii) **Convexity and Smoothness:** $f_i$ *is convex and 1-smooth.*

## I.2 Proof of Corollary 4

Now, we can turn to the proof of Corollary 4.

*Proof of Corollary 4.* First, for the with-replacement setting, let $\tau$ be a random with-replacement ordering, and let $x_1, \ldots, x_T$ denote the iterates generated by Eq. (5). By Lemma 10, these iterates coincide with those of SGD initialized at $w_1$, using step size $\eta = c$, on the sequence of loss functions $f_{\tau(1)}, \ldots, f_{\tau(T)}$, where

$$f_i(x) \coloneqq \frac{1}{2} \|A_i^+ A_i (x - x^\star)\|^2.$$

Moreover, by Lemma 11, for any $x \in \mathbb{R}^d$, $F$ satisfies:

$$F(x) \le R^2 \mathbb{E}_{i \sim \text{Unif}([m])} f_i(x).$$

Thus, it suffices to analyze last-iterate convergence of with-replacement SGD on $F$. Again by Lemma 11, each $f_i$ is 1-smooth. so we may invoke Theorem 1. This theorem guarantees that after $T \ge 1$ gradient steps with step size $\eta = c$, it holds that,

$$\mathbb{E}_{i \sim \text{Unif}([m])} f_i(x) \le \frac{3\|x_1 - x^\star\|^2}{\eta T^{1 - c\beta/2}}.$$

Combining this with the earlier inequality yields:

$$\mathbb{E}_\tau F(x_T) \le \frac{3 R^2 \|x^\star\|^2}{\eta T^{1 - c\beta/2}}.$$

For the without-replacement case, the proof is analogous and is obtained by using Theorem 3 instead of Theorem 1 (the constant in the bound will be 13 instead of 3). $\qquad \square$

## I.3 Proof of Corollary 5

We use the following lemma from Evron et al. [2025].

**Lemma 12** (Reduction 1 in Evron et al. [2025]). *In the realizable case, under any ordering $\tau$, continual linear regression learned to convergence is equivalent to the block Kaczmarz method with $c = 1$. That is, the iterates $x_1, \ldots, x_T$ of Eqs. (5) and (6) coincide.*

Now, we can turn to the proof of Corollary 4. We prove for $x_T$ and the bounds for $x_{T+1}$ can be obtained by another sampling from the distribution.

*Proof of Corollary 5.* By Lemma 12, for $c = 1$, the iterates of Eqs. (5) and (6) concide. Then, by Corollary 4, we get that in both regimes, with and without replacement, it holds that,

$$\mathbb{E}_\tau F(x_T) \leq \frac{13R^2 \|x^\star\|^2}{\sqrt{T}}.$$

For the forgetting, in the with-replacement case, by Lemma 3, we get that,

$$\mathbb{E}_\tau F_\tau(x_T) \leq \frac{30R^2 \|x^\star\|^2}{\sqrt{T}}.$$

In the without-replacement case, by Lemmas 8 and 9, we get that,

$$\mathbb{E}_\tau F_\tau(x_T) \leq \frac{10R^2 \|x^\star\|^2}{\sqrt{T}}.$$

$\square$

