# OpenReview forum: "Fast Last-Iterate Convergence of SGD in the Smooth Interpolation Regime"
_NeurIPS.cc/2025/Conference — NeurIPS 2025 poster_

### Official Review · Reviewer_4zP2 · 2025-06-08

**Clarity:** 3
**Significance:** 3
**Originality:** 3
**Rating:** 5
**Confidence:** 3

**Summary:**

This paper studies the last-iterate convergence of SGD. For the general smooth convex setting with optimally tuned learning rates, it obtains an improved instance dependent bound which replaces the uniform noise bound in the previous work by the variance at optimum. This directly implies an improved bound for the realizable setting, that goes beyond the OLS regime. For the realizable setting with the greedy stepsize $1/\beta$, it obtains a strictly improved convergence guarantee which has a faster rate and goes beyond OLS.

**Questions:**

I wonder if last-iterate convergence guarantees can be potentially obtained for the agnostic setting with the natural $\eta=1/\beta$ choice. If it's not theoretically impossible, what's the main technical difficulty?

**Ethical Concerns:**

["NO or VERY MINOR ethics concerns only"]

**Final Justification:**

Author's response addressed my question, and I will maintain my score.

**Limitations:**

Yes.

**Quality:**

3

**Strengths And Weaknesses:**

**Strengths:** studying the last-iterate convergence of SGD is an important problem both theoretically and practically. This paper provides solid advancement on this problem. The technical framework and novelty are well explained: the proof outline clearly describes the roadmap, making it easier for the readers to tell which parts (step 0,1,3) mostly follow from the literature and which (step 2,4,5) are novel technical contributions. In my opinion, the overall approach still lies in the classic framework, but the technical contributions are solid and necessary for obtaining the improved rates. Hence I recommend accept.

**Weaknesses:** the improvement is mainly in the realizable setting with the greedy stepsize. For the general setting, the improvement is somewhat minor, but it's still a strict improvement since the algorithm requires no knowledge of the variance at optimum.

---

> ### Author Rebuttal · Authors · 2025-07-28
>
> Thank you for reviewing our work and for your support of our work. Please see below our responses to the points you raised.
>
> > “the improvement is mainly in the realizable setting with the greedy stepsize. For the general setting, the improvement is somewhat minor, but it's still a strict improvement since the algorithm requires no knowledge of the variance at optimum.”
>
> In the realizable setting, our improvement in fact holds for any stepsize asymptotically larger than $1/\sqrt{T}$ and not only for the greedy stepsize $\eta=1/\beta$. Beyond the realizable setting, our work provides for the first time the (nearly) optimal last-iterate convergence rate of $1/T+\sigma_\star/\sqrt{T}$ in the general convex and smooth case. This rate was previously known only in the special case of ordinary least squares (Varre et al [32]). We view this as a significant result. You are correct to note that none of these results require knowledge of the variance at optimum.
>
> > “I wonder if last-iterate convergence guarantees can be potentially obtained for the agnostic setting with the natural $\eta=1/\beta$ choice. If it's not theoretically impossible, what's the main technical difficulty?”
>
> In the agnostic case (which is, to our understanding, the non-realizable case with $\sigma_\star >> 0$), the last-iterate sub-optimality of SGD $\eta=1/\beta$ does not diminish to zero. The best one can hope for is a rate of $O(1/T+\sigma_\star^2)$, and as we discuss in the paper, there is indeed still a gap between our guarantees and the best known lower-bounds in this setting.
>
> From a technical perspective, the main difficulty in improving the bounds is that our current analysis technique is sensitive to multiplicative numerical constants along the analysis, as they carry over to constants in the exponents and therefore translate to polynomial differences in rates. We do not know if a new and innovative new technique is necessary for closing the gap.

---

> > ### Comment · Reviewer_4zP2 · 2025-08-05
> >
> > Thank you for the response. For the last-iterate convergence rate of $1/T+\sigma_\star/\sqrt{T}$ in the general convex and smooth case, its improvement over Liu and Zhou [22] is an instance-dependent noise norm bound, instead of dependence on $T$. In addition, according to table 1, the result of Varre et al [32] is a different bound in a different setting. In all, I will maintain my score.

---

> > > ### Author Response · Authors · 2025-08-05
> > >
> > > Thank you for your message and your support. We would like to emphasize a few quick points:
> > >
> > > - The improvement over Liu and Zhou [22] is indeed not in terms of the dependence on $T$, as the rate there was already optimal for small ($T$-dependent) stepsizes. We improve the dependence on the maximal noise variance $\sigma$ in their result to a dependence on the noise variance at optimum $\sigma_*$, which yields last-iterate bounds that match the state of the art for the average iterate of SGD. There is a significant difference between the two types of bounds: e.g., when $\sigma=0$ the Liu and Zhou setting degenerates to fully deterministic, non-stochastic optimization, whereas in our case $\sigma_* = 0$ corresponds to the stochastic interpolation regime.
> > >
> > > - The result of Varre et al [32] was in the setting of ordinary least squares (OLS), which is a *special case* of smooth convex optimization; our bound for $\eta = 1/(\beta\log{T})$ precisely matches theirs (up to constants) and generalizes it significantly to smooth convex stochastic problems.
> > >
> > > - Finally, just to reiterate: the way we see it, one of our main novelties (and motivations for this work) is in establishing last-iterate rates for large (constant) stepsizes, and particularly for $\eta=1/\beta$ being crucial for the applications we consider. This regime was not addressed by either Liu and Zhou [22] or Varre et al [32].

---

### Official Review · Reviewer_ixqP · 2025-06-30

**Clarity:** 3
**Significance:** 2
**Originality:** 2
**Rating:** 4
**Confidence:** 3

**Summary:**

The paper studies population convergence guarantees of stochastic gradient descent (SGD) for smooth convex objectives in the interpolation (low noise) regime.

- The paper establishes a convergence rate of $\widetilde{O}(1/(\eta T^{1-\beta\eta/2}) + \eta T^{\beta\eta/2}\sigma_{\star}^2)$ for the last iterate of SGD with step-size $\eta \leq 1 /\beta$ on $\beta$-smooth convex loss functions, where $\sigma_{\star}^2$ is the variance of the stochastic gradients at the optimum. With a well-tuned stepsize, this translates to a $\widetilde{O}(1/T + \sigma_{\star}/\sqrt{T})$ rate.

- The paper further extends the analysis to without-replacement SGD, block Kaczmarz method, and continual linear regression, leading to some better convergence rates.

**Questions:**

- See the weakness part.
- Is there any connections between the Zamani and Glineur technique with [1, Theorem 2]?
- How can the result from Srebro et al. (2010), which uses an approximation error rather than $\sigma^*$, be refined to obtain the error bound presented in Table 1.


[1] Francesco Orabona. Last iterate of sgd converges (even in unbounded domains), 2020. URL
https://parameterfree.com/2020/08/07/last-iterate-of-sgd-converges-even-in-unbounded-domains/.

**Ethical Concerns:**

["NO or VERY MINOR ethics concerns only"]

**Final Justification:**

I keep my positive support.

**Limitations:**

probably yes

**Quality:**

3

**Strengths And Weaknesses:**

**Strengths**

- The paper derives a $\widetilde{O}(1/T + \sigma_{\star}/\sqrt{T})$ rate for the last iterate of SGD with a well tuned step-size on smooth convex loss functions. It improves previous result by replacing the global noise variance bound with the variance at the optimum, or under the same set-up, it extends convergence guarantees for averaged iterates  to the last iterate of SGD, matching (up to logarithmic factors) the optimal rates achieved by averaging.

- The paper further extends the analysis to other settings.

- The paper is in general clearly written.

**Weakness**

- I agree with that  deriving convergence results for the last iterate, rather than the averages, of SGD is interesting. Given that the connection between the last iterate and averages has been extensively explored in the literature, the contribution feels somewhat incremental to me.
- Could the theoretical results of this paper offer practical insights into the numerical experiments? For instance, Theorem 1 suggests that the step-size in case (ii) should outperform the choice in (i). Is this actually observed in practical applications or numerical simulations?
- While the paper focuses on the convex setting—a reasonable starting point for theoretical analysis—this is somewhat limiting given the prevalence and importance of nonconvex optimization in modern machine learning applications.
-  As the authors acknowledge, when $\eta = {1\over \beta}$, the convergence rate is $1/\sqrt{T}$ in the interpolating regime, which falls short of the optimal $1/T$ rate achievable with averaged SGD.  Moreover, what is the theoretical motivation for considering this particular step-size setting?

---

> ### Author Rebuttal · Authors · 2025-07-28
>
> Thank you for your thoughtful review—we hope that the below will successfully address all your concerns, in which case we hope you could acknowledge that by updating your score. If any concern still remains however, we will be glad to try and clarify further in the discussion.
>
> > “I agree with that deriving convergence results for the last iterate, rather than the averages, of SGD is interesting. Given that the connection between the last iterate and averages has been extensively explored in the literature, the contribution feels somewhat incremental to me.”
>
> We actually believe the fact that a lot of effort has been put into last-iterate analyses over the past decade, while still leaving this fundamental gap open, only emphasizes the significance of our results. In addition, obtaining the new result required new technical innovations, as described in Section 3.1, beyond the techniques already explored in the literature.
>
> > “... Theorem 1 suggests that the step-size in case (ii) should outperform the choice in (i). Is this actually observed in practical applications or numerical simulations?”
>
> The gap between cases (i) and (ii) is intriguing. From a theoretical perspective, it remains an open question whether this gap is real, as it is possible that case (i) already achieves the optimal $1/T$ rate. And in practice, even if the gap exists, observing it empirically would be challenging since the number of steps $T$ is usually fixed, making it difficult to distinguish between a $\Theta(1/\beta)$ stepsize and a $\Theta(1/(\beta\log{T}))$ stepsize.
>
> > “While the paper focuses on the convex setting---a reasonable starting point for theoretical analysis---this is somewhat limiting given the prevalence and importance of nonconvex optimization in modern machine learning applications.”
>
> The fundamental convex case is the focus of the extensive literature on last-iterate guarantees of SGD, and we have yet to reach a full understanding of convergence and rates even in this case. And indeed, the progress we made in the convex case already required nontrivial technical innovations that will likely be crucial for further advances.
>
> As an aside, it is a folklore fact that meaningful last-iterate guarantees are generally not possible in the non-convex smooth setting, even without noise. (There are simple examples of smooth problems where the norm of the gradient actually increases with time, rather than decreases.)
>
> > “As the authors acknowledge, when $\eta=\frac{1}{\beta}$, the convergence rate is $1/\sqrt{T}$ in the interpolating regime, which falls short of the optimal $1/T$ rate achievable with averaged SGD. Moreover, what is the theoretical motivation for considering this particular step-size setting?”
>
> As we explain in the paragraph starting at line 40 and elaborate in Section 5, the choice $\eta=\frac{1}{\beta}$ is crucial in several applications of SGD related to continual learning and the Kaczmarz method, which were actually our main motivation for studying this problem. See Sections 5.2 and 5.3 for more details on these applications.
>
> > “Is there any connections between the Zamani and Glineur technique with [1, Theorem 2]?”
>
> Theorem 2 in [1] is based on the Shamir & Zhang analysis technique that, as we discuss in Section 3.1, was used in previous work that was only able to achieve a weaker $1/T^{1/4}$ rate for $\eta=1/\beta$, compared to the $1/\sqrt{T}$ we establish here.
>
> The Zamani and Glineur technique can be seen as an intricate adaptation of Shamir & Zhang, that instead of using the standard SGD potential analysis with relation to previous iterates of the algorithm, it uses such an analysis with respect to certain **convex combinations** of previous iterates.
>
> > “How can the result from Srebro et al. (2010), which uses an approximation error rather than $\sigma_\star$, be refined to obtain the error bound presented in Table 1.”
>
> In Appendix I we provided the refined analysis you seem to allude to, which provides the error bound presented in Table 1. In addition, this analysis applies to a wider range of stepsizes (including $\eta=\frac{1}{\beta}$) compared to the original analysis of Srebro et al. (2010).

---

> > ### Comment · Reviewer_ixqP · 2025-08-05
> >
> > Thank you for your response.
> >
> > While some of my concerns remain unresolved (and, as the authors noted, certain aspects may not be fully addressable), I nevertheless keep the positive support.

---

> > > ### Author Response · Authors · 2025-08-05
> > >
> > > Thank you for your message and for the support - we would sincerely appreciate it if you could kindly let us know which of your concerns remains unresolved (it is not obvious to us from re-reading your review), so we could benefit from the review process and try to address those in our final version.

---

### Official Review · Reviewer_54Za · 2025-07-02

**Clarity:** 4
**Significance:** 4
**Originality:** 3
**Rating:** 6
**Confidence:** 4

**Summary:**

This paper provides a significantly improved last-iterate convergence analysis of SGD for smooth, convex functions in the interpolation and low noise regimes. This is the first result showing a $1/T$ last-iterate rate in the interpolation regime (previous results had achieved this rate only for ordinary least squares), and this is the first result showing a $1/\sqrt{T}$ last-iterate rate with a greedy stepsize $\eta = 1/\beta$ (where $\beta$ is the smoothness constant), where previous work had shown a $1/T^{1/4}$ rate and only for ordinary least squares.

**Questions:**

1. Can you explain what exactly is the low-noise regime? The introduction (line 62) says that $\sigma_* \ll 1$, but the statement of Theorem 2 only requires that $\sigma_* < \infty$. Assuming that low-noise refers to $\sigma_* < \infty$, why call this low-noise? Bounded variance (or bounded variance at optimum) is a common assumption in optimization, but is this condition often referred to as low-noise?
2. Liu and Zhou [22] have results for both in-expectation and high-probability. Do you think your techniques can be easily extended to allow for high-probability guarantees, assuming say sub-gaussian noise as they do?

**Ethical Concerns:**

["NO or VERY MINOR ethics concerns only"]

**Final Justification:**

After my initial review, I did not have any major concerns about the paper, just a few small questions which the authors answered with the rebuttal. I believe that this is a strong theoretical work, clearly worthy of publication.

**Limitations:**

Yes

**Paper Formatting Concerns:**

None.

**Quality:**

4

**Strengths And Weaknesses:**

Strengths:
1. This is an important problem: the last-iterate convergence of SGD (especially under interpolation) is fundamental for optimization with clear motivation for machine learning.
2. The improvement over previous work is clear. Better rates are established and in more general settings (see Table 1).
3. The paper is very well-written and clearly communicates the novelty in the analysis compared to previous work. I think the proof outline and proof itself (sections 3.1 and 3.2) are informative and transparent about the ideas.

Weaknesses:
1. The $1/T^{1/4}$ rate for greedy stepsize with interpolation does not match lower bounds ($1/T$) or upper bounds for the average iterate ($1/T$), though I think this is fine. The authors acknowledge this gap as an open problem (line 77) and I think this is gap is totally reasonable: you can't solve everything in one paper.

---

> ### Author Rebuttal · Authors · 2025-07-28
>
> Thank you for reviewing our work and providing us with your strong support. It is highly appreciated and encouraging. Please see below our responses to the points you raised.
>
> > “Can you explain what exactly is the low-noise regime? The introduction (line 62) says that $\sigma_\star \ll 1$, but the statement of Theorem 2 only requires that $\sigma_\star < \infty$. … Bounded variance (or bounded variance at optimum) is a common assumption in optimization, but is this condition often referred to as low-noise?”
>
> You are correct to note that our results only require an assumption of bounded variance at the optimum, leading to a rate of $\widetilde{O}(1/T + \sigma_\star / \sqrt{T})$ for an optimally tuned stepsize. The term “low-noise” in this context (originally coined by Srebro et al (2010), to our knowledge) refers to cases where $\sigma_\star = o(1)$ (in terms of $T$), in which case a convergence rate strictly better than $1/\sqrt{T}$ rate is possible. Thanks for this comment - we will try to make this terminology clearer in the final version.
>
> > “Liu and Zhou [22] have results for both in-expectation and high-probability. Do you think your techniques can be easily extended to allow for high-probability guarantees, assuming say sub-gaussian noise as they do?”
>
> This is a very interesting question we have yet to thoroughly explore. While the results of Liu and Zhou [22] suggest that high-probability last-iterates bounds of this form are achievable, the interplay between the concentration and the weaker noise assumption (bounded variance at the optimum instead of a global variance bound) seems to lead to several technical challenges that we do not yet know how to circumvent.

---

> > ### Comment · Reviewer_54Za · 2025-08-04
> >
> > Thank you for your response. I maintain my positive opinion of this submission and I will keep my score.

---

> > > ### Author Response · Authors · 2025-08-05
> > >
> > > Thanks again for your review and support!

---

### Decision · Program_Chairs · 2025-09-17

**Decision:**

Accept (poster)

**Comment:**

The paper studies convergence of the last iterate of SGD applied to $\beta$-smooth convex functions, in the "interpolation" regime (i.e., where the variance $\sigma_*$ at the optimum is very small or zero). The results are also extended to sampling without replacement and continual learning settings. The paper obtains a bound on the optimality gap that allows for large step sizes, up to $1/\beta,$ for arbitrary value of $\sigma_*.$ For tuned fixed step size, this bound recovers known bounds, which are of the order $1/T + \sigma_*/\sqrt{T}.$ Importantly, unlike prior results (which necessarily forced the step size to be sufficiently small and dependent on $T$), this work allows for *large* step sizes and provides informative bounds even when the step size is as large as $1/\beta$, assuming $\sigma_*$ is small compared to $1/T$ (e.g., $\sigma_* = O(1/T)$).

The results are interesting, considering there has been a lot of recent interest on this topic and none of the prior results could provide informative bounds for large step sizes. The reviews were in uniform agreement about the paper's strength and thus I recommend acceptance.